# A New Rejection Sampling Approach to $k$-`means` **++ with Improved Tradeoffs**

## Abstract

The $k$-means++ seeding algorithm (Arthur & Vassilvitskii, 2007) is widely used in practice for the $k$-means clustering problem where the goal is to cluster a dataset $\mathcal{X} \subset \mathbb{R}^d$ into $k$ clusters. The popularity of this algorithm is due to its simplicity and provable guarantee of being $O(\log k)$ competitive with the optimal solution in expectation. However, its running time is $O(|\mathcal{X}|kd)$, making it expensive for large datasets. In this work, we present a simple and effective rejection sampling based approach for speeding up $k$-means++. Our first method runs in time $\tilde{O}(\texttt{nnz}(\mathcal{X}) + \beta k^2 d)$ while still being $O(\log k)$ competitive in expectation. Here, $\beta$ is a parameter which is the ratio of the variance of the dataset to the optimal $k$-means cost in expectation and $\tilde{O}$ hides logarithmic factors in $k$ and $|\mathcal{X}|$. Our second method presents a new trade-off between computational cost and solution quality. It incurs an additional scale-invariant factor of $k^{-\Omega(m/\beta)} \operatorname{Var}(\mathcal{X})$ in addition to the $O(\log k)$ guarantee of $k$-means++ improving upon the result of (Bachem et al., 2016a) who get an additional factor of $m^{-1} \operatorname{Var}(\mathcal{X})$ while still running in time $\tilde{O}(\texttt{nnz}(\mathcal{X}) + mk^2 d)$. We perform extensive empirical evaluations to validate our theoretical results and to show the effectiveness of our approach on real datasets.

## 1. Introduction

Data clustering has numerous applications in data processing and is one of the classic problems in unsupervised machine learning. Its formulation as the $k$-means problem is defined as: given a data set $\mathcal{X} \subset \mathbb{R}^d$ and a positive integer $k$ representing the number of clusters into which the dataset is to be partitioned, find a set $C \subset \mathbb{R}^d$ of $k$ centers such that

[1]Anonymous Institution, Anonymous City, Anonymous Region, Anonymous Country. Correspondence to: Anonymous Author <anon.email@domain.com>.

Preliminary work. Under review by the International Conference on Machine Learning (ICML). Do not distribute.

the following objective or cost function is minimized :

$$\Delta(\mathcal{X}, C) \coloneqq \sum_{x \in \mathcal{X}} \min_{c \in C} \|x - c\|^2$$

The set $C$ implicitly defines a partition of $\mathcal{X}$ based on the closest center from $C$. A set of centers which achieve the minimum $k$-means cost is denoted by $\texttt{OPT}_k = \{c_1^*, \ldots, c_k^*\}$. We shall be using the shorthand $\Delta_k(\mathcal{X}) \coloneqq \Delta(\mathcal{X}, \texttt{OPT}_k)$ to refer to the optimal $k$-means cost.

**Background on the $k$-means problem**. On the hardness front, solving the $k$-means problem exactly is known to be NP-hard (Dasgupta, 2008), even when the data points are restricted to lie in a plane (Mahajan et al., 2009). Moreover, there exists a constant $c > 1$ such that it is NP-hard to solve the $c$-approximate version of $k$-means where we are allowed to output cluster centers $C$ such that $\Delta(\mathcal{X}, C) \leq c\Delta_k(\mathcal{X})$ (Awasthi et al., 2015; Lee et al., 2017; Cohen-Addad & C.S., 2019) . On the algorithmic front, a significant amount of effort has been put into designing algorithms for $k$-means that have strong theoretical guarantees. These include, for example, the constant factor approximation results of (Jain & Vazirani, 2001; Kanungo et al., 2002; Ahmadian et al., 2020; Cohen-Addad et al., 2022) and the $(1 + \varepsilon)$ approximation schemes of (Kumar et al., 2010; Jaiswal et al., 2014; 2015; Cohen-Addad, 2018; Friggstad et al., 2019; Cohen-Addad et al., 2019; Bhattacharya et al., 2020) which have exponential dependence on one or more of $\varepsilon^{-1}, k$ or $d$. While these works provide important insights into the structure of the $k$-means problem, they are seldom used in practice due to their slow speed. Indeed, one of the most popular heuristics used in practice (Wu et al., 2008) is Lloyd's iterations (Lloyd, 1982), also referred to as the $k$-means method. It starts off with an initial set of centers [1] and iteratively refines the solution. This hill-climbing approach may get stuck in local minima and provide arbitrarily bad clusterings even for fixed $n$ and $k$ (Dasgupta, 2003; Har-Peled & Sadri, 2005; Arthur & Vassilvitskii, 2006a;b).

$k$-means **++ and $D^2$-sampling.** Usually, Lloyd's iterations are preceded by the $k$-means ++ seeding introduced in (Arthur & Vassilvitskii, 2007). Even though the $k$-means++ algorithm is the Lloyd's iterations preceded by $k$-means++

---

[1]This is commonly known as *seeding*. A simple seeding method is to arbitrarily pick $k$ points from $\mathcal{X}$.

seeding, it is common to refer to the seeding procedure as $k$-means++. We follow this in the remaining discussion. $k$-means ++ is a fast sampling-based approach. Starting with a randomly chosen center $S = \{c_1\}$, a new point $x \in \mathcal{X}$ is chosen as the next center with probability proportional to $\Delta(\{x\}, S)$ in each iteration. This is commonly referred to as $D^2$-sampling. The centers generated by this seeding method are guaranteed to be $O(\log k)$ competitive with the optimal solution in expectation. Thus, $k$-means ++ provides the best of both worlds : theory and practice and unsurprisingly, a lot of work has been done on it. This includes extending it to the distributed setting (Bahmani et al., 2012) and the streaming setting (Ailon et al., 2009; Ackermann et al., 2012). Furthermore, several results on coreset constructions [2] are inspired by or rely on the theoretical guarantees of $k$-means ++. Recently, it was shown that appending $k$-means ++ with a sufficiently large number of local search steps (Lattanzi & Sohler, 2019; Choo et al., 2020) can lead to $O(1)$ competitive solutions.

A downside of $k$-means ++ is that its $\Theta(nkd)$ computational complexity becomes impractical on large datasets. Various approaches (Bachem et al., 2016a;b; Cohen-Addad et al., 2020; Charikar et al., 2023) have been presented to speed up $k$-means ++ with varying trade-offs, and our work also falls into this category. A detailed discussion about the position of our approach in the literature is presented in Section 2.3. We also include Table 1 as a summary for reference.

## 2. Our Results

In this section, we present a high level discussion of our results, contributions and their significance.

**Improved tradeoffs.** Our main technical contribution is a novel simple yet fast algorithm based on rejection sampling with an improved trade-off between the computational cost and solution quality for $k$-means ++ in the Euclidean metric. A description is given in Algorithm 1. We state our result formally below.

**Theorem 2.1.** *(Main Theorem) Let $m \in \mathbb{N}$ be a parameter and $k \in \mathbb{N}$ be the number of clusters. Let $\mathcal{X} \subset \mathbb{R}^d$ be any dataset of $n$ points and $S$ be the output of* RS-k-means++ *$(\mathcal{X}, k, m')$ where $m' = cm \ln k$ for some constant $c > 1$. Then the following guarantee holds :*

$$\mathbb{E}[\Delta(\mathcal{X}, S)] \leq 8(\ln k + 2)\Delta_k(\mathcal{X}) + \frac{6k}{k^{\frac{cm}{2\beta(\mathcal{X})}} - 1}\Delta_1(\mathcal{X})$$

*Here $\beta(\mathcal{X})$ [3] is a parameter such that $\mathbb{E}[\beta(\mathcal{X})] = \frac{\Delta_1(\mathcal{X})}{\Delta_k(\mathcal{X})}$. Moreover, the computational cost of the algorithm includes*

---

[2]See, for example (Bachem et al., 2017a; Feldman, 2020) and the extensive references cited therein.

[3]As can be seen from the description, the value of $\beta(\mathcal{X})$ is not needed to be known by our algorithm

*a single-time preprocessing cost of $\tilde{O}(\text{nnz}(\mathcal{X}))$ [4], with the cost of performing a single clustering being $O(mk^2 d \log k)$.*

To the best of our knowledge, such trade-offs were not known before this work. The approximation guarantee can be seen to be composed of two terms. The first term is the standard $O(\log k)$ guarantee of $k$-means ++, while the second term can be thought of as an additive, scale-invariant term representing the variance of the dataset. Note that as $m$ grows, the second term diminishes rapidly. Indeed, this exponentially decreasing dependence of $k^{-\Omega(m/\beta(\mathcal{X}))}$ improves on a similar result by (Bachem et al., 2016a) who instead get a linearly decreasing dependence of $O(1/m)$, although through a significantly different approach.

**Correct number of iterations.** Whenever we have such trade-offs, a natural question to ask is : for which value of $m$ can we get $O(\log k)$ competitive solutions like those of $k$-means ++ ? For example, we require $m = \Omega\left(\frac{\Delta_1(\mathcal{X})}{\Delta_k(\mathcal{X})}\right)$ in (Bachem et al., 2016a)'s algorithm. But this means that we would some how need to get an estimate for $\Delta_k(\mathcal{X})$, which involves solving the $k$-means problem itself ! Fortunately, Algorithm 1 can *"discover"* the value of $\beta(\mathcal{X})$ as it executes. We state this as follows :

**Theorem 2.2.** *Let $\epsilon \in (0, 1)$ and $k \in \mathbb{N}$ be the number of clusters. Let $\mathcal{X} \subset \mathbb{R}^d$ be any dataset of $n$ points and $S$ be the output of* RS-k-means++ *$(\mathcal{X}, k, \infty)$. Then the following guarantee holds :*

$$\mathbb{E}[\Delta(\mathcal{X}, S)] \leq 8(\ln k + 2)\Delta_k(\mathcal{X})$$

*Moreover, the computational cost of the algorithm includes a single-time preprocessing cost of $\tilde{O}(\text{nnz}(\mathcal{X}))$ with the cost of performing a single clustering being bounded by $O(\beta(\mathcal{X})k^2 d \log(k/\epsilon))$ with probability atleast $1 - \epsilon$. Here, $\beta(\mathcal{X})$ is a parameter such that $\mathbb{E}[\beta(\mathcal{X})] = \frac{\Delta_1(\mathcal{X})}{\Delta_k(\mathcal{X})}$.*

**Experimental results.** We evaluate our algorithms experimentally on several data sets as described in Section 5.

### 2.1. Overview of Our Techniques

**Algorithm.** Our main algorithm is outlined in Algorithm 1. It consists of a light-weight pre-processing step followed by choosing new centers according to the procedure D²-sample. This procedure consists of two parts : the first part is a rejection sampling loop, which generates samples distributed according to the $D^2$ distribution using samples generated from a specific distribution which is *easy to sample from*, being setup during the pre-processing itself. In case no sample is generated in $m$ iterations, the

---

[4]$\text{nnz}(\mathcal{X})$ represents the *number of non zero entries* in the dataset $\mathcal{X}$. When $\mathcal{X}$ is sparse, this can be much smaller than $nd$.

second part consists of choosing the next center uniformly at random.

**Proof intuition.** To analyze the expected solution quality of RS-k-means++, we study a variant of `k-means++` which we call $\delta$-k-means++ . In this variant , instead of sampling the next center from the $D^2$ distribution $p(x) = \frac{\Delta(x,S)}{\Delta(\mathcal{X},S)}$, we sample from a different distribution defined by

$$p'(x) = (1-\delta)\frac{\Delta(x,S)}{\Delta(\mathcal{X},S)} + \delta\frac{1}{|\mathcal{X}|}$$

The parameter $\delta$ can be thought of as representing the probability that `sampled = False` after the **repeat** loop is executed. If this event happens, we choose a center uniformly at random. Consider the case when $\delta = 0$ : this means that we get $O(\log k)$ competitive solutions since we sample exactly from the $D^2$ distribution. Now consider the case when $\delta = 1$. This corresponds to choosing all centers uniformly at random. It can be seen [5] that in this case, we have $\mathbb{E}[\Delta(\mathcal{X},S)] \leq 2\Delta_1(\mathcal{X})$. So, we expect that $\delta \in (0,1)$ leads to a trade-off between these two terms. The technical analysis of error propagation due to the use of a slightly perturbed distribution may be of independent interest.

---

**Algorithm 1** RS-k-means++ $(\mathcal{X}, k, m)$

---

**Input :** dataset $\mathcal{X} \subset \mathbb{R}^d$, number of clusters $k \in \mathbb{N}$ and the upper bound on number of iterations $m \in \mathbb{N}$
**Output :** $S = \{c_1, \ldots, c_k\} \subset \mathcal{X}$

  1: `preprocess`$(\mathcal{X})$
  2: Choose $c_1 \in \mathcal{X}$ uniformly at random and set $S \leftarrow \{c_1\}$
  3: **for** $i \in \{2, \ldots, k\}$ **do**
  4:    $c_i \leftarrow$ D$^2$-`sample`$(\mathcal{X}, S, m)$
  5:    $S \leftarrow S \cup \{c_i\}$
  6: **end for**
  7: **return** $S$

---

**Procedure 2** `preprocess`$(\mathcal{X})$

---

**Input :** dataset $\mathcal{X} \subset \mathbb{R}^d$
**Ensure :** $\mathcal{X}$ is centered

  1: Compute the mean $\mu(\mathcal{X})$ of the dataset $\mathcal{X}$ and perform $x \leftarrow x - \mu(\mathcal{X})$ for every $x \in \mathcal{X}$
  2: Setup the sample and query access data structure to enable sampling from the distribution $D_{\mathcal{X}}(x) = \frac{\|x\|^2}{\|\mathcal{X}\|^2}$

---

### 2.2. Advantages of our approach

**Fast data updates.** Rejection sampling essentially involves converting samples from a distribution which is

---

[5]The cost considering all centers is upper bounded by the cost considering only the first center. Since it is chosen uniformly at random , we can use Lemma 3.1 of (Arthur & Vassilvitskii, 2007).

---

**Procedure 3** D$^2$-`sample`$(\mathcal{X}, S, m)$

---

**Input :** dataset $\mathcal{X} \subset \mathbb{R}^d$, currently chosen centers $S \subset \mathcal{X}$ and upper bound on number of iterations $m \in \mathbb{N}$
**Output :** next center $c \in \mathcal{X}$

  1: `iter` $\leftarrow 0$ and `sampled` $\leftarrow$ `False`
  2: **repeat**
  3:    `iter` $\leftarrow$ `iter` $+ 1$
  4:    $r \sim [0, 1]$
  5:    Choose $x \in \mathcal{X}$ with probability $\frac{\|x\|^2 + \|c_1\|^2}{\|\mathcal{X}\|^2 + |\mathcal{X}|\|c_1\|^2}$
  6:    Compute $\rho(x) = \frac{1}{2}\frac{\Delta(x,S)}{\|x\|^2 + \|c_1\|^2}$
  7:    **if** $r \leq \rho(x)$ **then**
  8:       Set $c$ to be $x$ and `sampled` $=$ `True`
  9:    **end if**
 10: **until** `sampled` $=$ `True` or `iter` $> m$
 11: **if** `sampled` $=$ `False` **then**
 12:    Choose $c \in \mathcal{X}$ uniformly at random
 13: **end if**
 14: **return** $c$

---

*"easy to sample from"* to a required distribution. The single time pre-processing sets up a simple binary tree data structure [6] for sampling from an appropriate distribution. This structure supports addition and update of a data point in $O(\log|\mathcal{X}|)$ time while taking up only $O(\text{nnz}(\mathcal{X}))$ additional space. The details are given in Section A.2.

**Parallel setting.** The simplicity of our approach extends easily to parallel and distributed settings. We briefly discuss implementing the procedure D$^2$-`sample` in such settings. We assume that the dataset $\mathcal{X}$ is on a single machine which has $M$ cores. Suppose that the probability that a sample is output in a single round of the **repeat** loop is $p$. Recall that we have $p \geq \frac{\Delta_k}{2\Delta_1}$. The expected number of rounds that one must wait for a sample to be generated is atmost $2\Delta_1/\Delta_k$. Also notice that each round is independent of other rounds. So we can utilize all $M$ cores to perform rejection sampling until one of them outputs a sample. Hence, the probability that a sample is generated in a round now becomes $1 - (1-p)^M \geq 1 - e^{-pM}$. Hence the number of rounds needed to get a sample is atmost $\frac{e^{pM}}{e^{pM}-1}$ in expectation, which decreases drastically as $M$ increases.

### 2.3. Comparison with Related Work

In this section we compare our results for $k$-means ++ with other fast implementations having theoretical guarantees.

**MCMC methods.** The line of work (Bachem et al., 2016b;a) uses the Monte-Carlo-Markov-Chain based Metropolis-Hastings algorithm (Hastings, 1970) to approx-

---

[6]We were inspired by (Tang, 2019) which introduced a randomized linear algebra based framework for efficient simulation of *quantum machine learning* algorithms.

imate the $D^2$-distribution in $k$-means ++. This involves setting up a markov chain of length $m$ to generate samples from the $D^2$ distribution $p(\cdot)$ using samples from a proposal distribution $q(\cdot)$. (Bachem et al., 2016b) used $q(\cdot)$ as the uniform distribution. To bound the solution quality of their method, they introduce the following parameters :

$$\alpha(\mathcal{X}) := \max_{x \in \mathcal{X}} \frac{\Delta(x, \mu(\mathcal{X}))}{\Delta_1(\mathcal{X})} \quad \beta(\mathcal{X}) := \frac{\Delta_1(\mathcal{X})}{\Delta_k(\mathcal{X})},$$

and show that $\alpha(\mathcal{X}) \in O(\log^2 n)$ and $\beta(\mathcal{X}) \in O(k)$ under some assumptions on the data distribution that is natural, but NP-hard to check. By doing so, they bound the required chain length $m \in O(\alpha(\mathcal{X})\beta(X) \log k\beta(\mathcal{X})) \in O(k^3 d \log^2 n \log k)$ to achieve $O(\log k)$ competitive solutions. This was improved upon by (Bachem et al., 2016a) by using a more suitable proposal distribution which needs $O(nd)$ pre-computation time. By doing so, they get rid of dependence on $\alpha(\mathcal{X})$ while showing a tradeoff between computational cost and approximation guarantee (see Table 1) without any data assumptions. They incur an additional $O(1/m)\Delta_1(\mathcal{X})$ error for a runtime $\in O(mk^2 d \log k)$. Our rejection sampling approach has the advantage of being independent of $\alpha(\mathcal{X})$, providing a stronger guarantee with only $k^{-\Omega\left(\frac{m}{\beta(\mathcal{X})}\right)}\Delta_1(\mathcal{X})$ additive error and being easy to extend to the parallel setting. On the other hand, MCMC methods are generally viewed to be inherently sequential [7].

**Tree embeddings and ANNS.** (Cohen-Addad et al., 2020) introduced an algorithmically sophisticated approach to speeding up $k$-means ++, focusing on the large $k$ regime. They use `MultiTree` embeddings with $O(d)$ expected distance distortions to update the $D^2$ distribution efficiently. They then use locality-sensitive hashing-based data structures for approximate nearest neighbor search to speed up their algorithm. This adds a significant layer of complexity in implementation. Their runtime also depends on the aspect ratio $\eta$, which may be quite large in case there are points in the dataset which are very close to each other. It has better dependence on $k$ but additional $n^{O(1)}, \log^{O(1)} \eta$ factors and cubic dependence on $d$ [8]. Moreover, their algorithm is advantageous only for large $k \sim 10^3$. Note that they also use rejection sampling to take into account distance distortions, which is different from our use of rejection sampling. Our approach provides improved trade-offs while being simple.

**1-D projections.** (Charikar et al., 2023) proposed an efficient method to perform the $k$-means ++ seeding in 1

dimension in $O(n \log n)$ time with high probability. For a general $d$-dimensional dataset, they first project it on a randomly chosen $d$- dimensional gaussian vector followed by an application of the 1-D method. This allows them to get an extremely fast runtime of $O(\text{nnz}(\mathcal{X}) + n \log n)$. However, they only get $O(k^4 \log k)$ competitive solutions, which shows up in their experimental evaluations as well. They show how to get $O(\log k)$ competitive solutions by using coresets, but end up with an additional high degree $O(k^5 d \log k \log(k \log k))$ [9] dependence. This may be restrictive even for moderate values of $k$, while our algorithm only has $O(k^2)$ dependence.

**Other related works.** (Bachem et al., 2017b) showed similar trade-offs for the $k$-means $\|$ algorithm of (Bahmani et al., 2012) in the distributed setting. They also get an additive scale-invariant factor in the approximation guarantee which diminishes with increase in the number of rounds and the oversampling factor of $k$-means $\|$. In contrast, we present a new rejection sampling based algorithm for $k$-means ++ with improved trade-offs. More recently, (Jaiswal & Shah, 2024) proposed an algorithm for performing the $k$-means ++ seeding in $\tilde{O}(nd + \eta^2 k^2 d)$ by using the framework of (Tang, 2019) through a data structure similar to the one used by us in the pre-processing step.

## 3. Preliminaries

For any two points $p, q \subset \mathbb{R}^d$, $\|p - q\|$ denotes their Euclidean distance. Throughout the paper, we denote the $d$ dimensional dataset to be clustered by $\mathcal{X} \subset \mathbb{R}^d$ with $|\mathcal{X}| = n$. For a set of points $\mathcal{P} \subset \mathbb{R}^d$, The number of non-zero elements in $\mathcal{P}$ is denoted by $\text{nnz}(\mathcal{P})$. Note that when all points in $\mathcal{P}$ are distinct, we have $|\mathcal{P}| \leq \text{nnz}(\mathcal{P})$. We define the *norm* of the set $\mathcal{P}$ to be the quantity $\|\mathcal{P}\| = \sqrt{\sum_{p \in \mathcal{P}} \|p\|^2}$. The k-means clustering cost of $\mathcal{P}$ with respect to a set of centers $C$ is denoted by :

$$\Delta(\mathcal{P}, C) = \sum_{p \in \mathcal{P}} \min_{c \in C} \|p - c\|^2$$

When either $\mathcal{P}$ or $C$ is a singleton set, we use expressions like $\Delta(p, C)$ or $\Delta(\mathcal{P}, c)$ instead of $\Delta(\{p\}, C)$ or $\Delta(\mathcal{P}, \{c\})$ respectively. The $D^2$ distribution over $\mathcal{P}$ with respect to $C$ is denoted by $D^2(\mathcal{P}, C)$ where the probability of a point $p \in \mathcal{P}$ being chosen is $\frac{\Delta(p,C)}{\Delta(\mathcal{P},C)}$. $D_{\mathcal{P}}$ denotes the distribution over $\mathcal{P}$ defined as $D_{\mathcal{P}}(p) = \frac{\|p\|^2}{\|\mathcal{P}\|^2}$ for each $p \in \mathcal{P}$. For a set $\mathcal{P}$ and a probability distribution $D$ over $\mathcal{P}$, $p \sim D$ denotes

---

[7] Note that the pre-processing step of (Bachem et al., 2016a) is easily parallelized.

[8] (Cohen-Addad et al., 2020) recommend using dimension reduction techniques such as the Johnson-Lindenstrauss transformation (Johnson & Lindenstrauss, 1984), which adds to the complexity of their approach.

[9] (Charikar et al., 2023) denote the size of the coreset as $s \in \Omega\left(\varepsilon^{-2} k \gamma d \log(k\gamma)\right)$ where $\gamma$ is the approximation ratio of the 1-d method i.e, $\gamma \in O(k^4 \log k)$ . This is only required for the theoretical guarantee of being $O(\log k)$ competitive to hold true. The coreset size can be treated as a hyper-paramter for trade-off between runtime and solution quality as well.

*Table 1.* Comparison of computational complexity and approximation guarantee of various approaches to speed up `k-means` ++. Here, $\Delta$ is the clustering cost for the centers returned by the algorithm and $\Delta_k$ is the optimal `k-means` cost

| APPROACH | COMP. COMPLEXITY | APPROX. GUARANTEE | REMARKS |
|---|---|---|---|
| (Bachem et al., 2016b) | $O(k^3 d \log^2 n \log k)$ | $\mathbb{E}[\Delta] \le 8(\ln k + 2)\Delta_k$ | The analysis only holds when the dataset satisfies certain assumptions which are NP-hard to check |
| (Bachem et al., 2016a) | $O(nd) + O(mk^2 d \log k)$ | $\mathbb{E}[\Delta] \le 8(\ln k + 2)\Delta_k + O\left(\frac{1}{m}\right)\Delta_1$ | $m$ is the markov chain length used |
| **Our** | $O(\texttt{nnz}(\mathcal{X})) + O(mk^2 d \log k)$ | $\mathbb{E}[\Delta] \le 8(\ln k + 2)\Delta_k + 6k^{-\Omega(m/\beta)}\Delta_1$ | $\texttt{nnz}(\mathcal{X})$ represents the input sparsity. The bound on number of iterations for rejection sampling is $O(m \log k)$. $\mathbb{E}[\beta] = \Delta_1/\Delta_k$ |
| (Cohen-Addad et al., 2020) | $O\left(n(d + \log n)\log(\eta d)\right) + O\left(\varepsilon^{-1} k d^3 \log \eta (n \log \eta)^{O(\varepsilon)}\right)$ | $\mathbb{E}[\Delta] \le 8\varepsilon^{-3}(\ln k + 2)\Delta_k$ | $\varepsilon \in (0,1)$ is a sufficiently small error factor for the LSH data structure . $\eta$ is the aspect ratio i.e, $\eta = \frac{\max_{x,y \in \mathcal{X}} \|x-y\|}{\min_{x,y \in \mathcal{X}} \|x-y\|}$ |
| (Charikar et al., 2023) | $O(\texttt{nnz}(\mathcal{X})) + O(n \log n)$ | $\mathbb{E}[\Delta] \le 51 k^4 (\ln k + 2)\Delta_k$ | $\texttt{nnz}(\mathcal{X})$ represents the input sparsity. The exact constant is upper bounded by $8\sqrt{24\sqrt{e}} \simeq 50.3$ |
| (Charikar et al., 2023) | $O(\texttt{nnz}(\mathcal{X})) + O(n \log n) + O(\varepsilon^{-2} k^5 d \log k \log(k \log k))$ | $\mathbb{E}[\Delta] \le 8(\ln k + 2)(1 + \varepsilon)\Delta_k$ | $\texttt{nnz}(\mathcal{X})$ represents the input sparsity. The high polynomial factor in $k$ is due to coreset constructions |

sampling a point $p \in \mathcal{P}$ with probability $D(p)$.

### 3.1. Data Dependent Parameter

The computation-cost vs. solution-quality trade-off of our algorithm depends on a data-dependent parameter which is bounded by $\beta(\mathcal{X}) := \Delta_1(\mathcal{X})/\Delta_k(\mathcal{X})$. Without any assumptions on $\mathcal{X}$, this parameter is unbounded (for example, if the data set had only $k$ points, then $\beta(\mathcal{X}) = \infty$, but as (Bachem et al., 2016b) point out, what is the point of clustering such a dataset if the solution is trivial ?). Indeed, if we assume that $\mathcal{X}$ is generated from some probability distribution over $\mathbb{R}^d$, this parameter becomes independent of $|\mathcal{X}|$, as $|\mathcal{X}|$ grows larger (Pollard, 1981). Moreover (Bachem et al., 2016b) showed that for a wide variety of commonly used distributions[10] $\beta(\mathcal{X}) \in O(k)$. In the experimental section, we shall also see that on many practical datasets, this parameter does not take on values which are prohibitively large [11].

## 4. Technical Overview

In this section, we describe the techniques used in our approach. Due to space constraints, the complete proofs of the results are deferred to the Appendix A, B. Instead, we try to provide intuition for our theoretical results.

---

[10]These include the uni-variate and multivariate Gaussian, the Exponential and the Laplace distributions along with their mixtures. For the exact assumptions made on the dataset, see section 5 of (Bachem et al., 2016b)

[11]Also see the estimated values this parameter for other datasets in Table 1 of (Bachem et al., 2016b)

Given a dataset $\mathcal{X} \subset \mathbb{R}^d$ and a set of already chosen centers $S \subset \mathcal{X}$, our goal is to obtain a sample from $\mathcal{X}$ according to the $D^2(\mathcal{X}, S)$ distribution. The main ingredient of our algorithm is a rejection sampling procedure which allows us convert samples from $D_{\mathcal{X}}$[12] to a sample from $D^2(\mathcal{X}, S)$. Why choose the starting point as $D_{\mathcal{X}}$ ? Because there exists a light-weight data structure that can efficiently generate samples from $D_{\mathcal{X}}$, which we describe next.

### 4.1. Preprocessing to sample from $D_{\mathcal{X}}$

Given $\mathcal{X} \subset \mathbb{R}^d$, consider the vector $v_{\mathcal{X}} \in \mathbb{R}^{|\mathcal{X}|}$ given by $v_{\mathcal{X}}(x) = \|x\|$. We will use a (complete) binary tree data structure to sample from $D_{\mathcal{X}}$. This data structure is also referred to as a *sample and query access* data structure. The leaves of the binary tree correspond to the entries of $v_{\mathcal{X}}$ and store weight $v_{\mathcal{X}}(x)^2$ along with the sign of $v_{\mathcal{X}}(x)$. The internal nodes also store a weight that is equal to the sum of weights of its children. To sample from $D_{\mathcal{X}}$, we traverse the tree, choosing either to go left or right at each node with probability proportional to the weight of its two children until reaching the leaves. This data structure similarly supports querying and updating the entries of $v_{\mathcal{X}}$. More details can be found in the appendix. For now, we state the following :

**Lemma 4.1.** *There exists a data structure that can be prepared in $\tilde{O}(\texttt{nnz}(\mathcal{X}))$ time and space, and enables generating a sample from $D_{\mathcal{X}}$ as well as updating an entry of $\mathcal{X}$ in $O(\log |\mathcal{X}|)$ time.*

---

[12]Recall that we defined $D_{\mathcal{X}}$ as the distribution over $\mathcal{X}$ with $D_{\mathcal{X}}(x) = \frac{\|x\|^2}{\|\mathcal{X}\|^2}$

### 4.2. Rejection Sampling

Now that we know how to sample from $D_\mathcal{X}$, let us describe the rejection sampling procedure.

**Definition 4.2.** Suppose $D_1$, $D_2$ define probability distributions over $\mathcal{X}$. The distribution $D_2$ is said to $\tau$-oversample $D_1$ for $\tau > 0$ if $D_1(x) \leq \tau D_2(x)$ for each $x \in \mathcal{X}$.

Let $D_1$, $D_2$ be probability distributions over $\mathcal{X}$ such that $D_2$ $\tau$-oversamples $D_1$. Suppose we have a collection of samples $x_1, x_2, \ldots$ from the distribution $D_2$. Consider the following strategy `RejectionSample`: iterate through samples $\{x_i\}_i$ and terminate if a sample $x_i$ is accepted with probability $\rho(x_i) = \frac{D_1(x_i)}{\tau D_2(x_i)}$.

It is not difficult to argue that an accepted sample comes from the distribution $D_1$. Moreover, for any $\varepsilon \in (0,1)$ it takes at most $\tau \ln(1/\varepsilon)$ samples from $D_2$ to accept a sample with probability at least $1 - \varepsilon$. Note that strategy does not need to know $\tau$ in advance (indeed, computing $\tau$ may be non trivial as we shall see), but only requires the ability to compute the quantity $\rho(x_i)$.

In the current form, our strategy does not have any control over the number of samples from $D_2$ which it may need to examine. However, a bound on the number of samples to be examined can be used if we are content with sampling from a slightly perturbed distribution. Suppose we have another distribution $D_3$ over $\mathcal{X}$. This time we are allowed to use samples coming from $D_2$ and $D_3$ and instead of a sample from $D_1$, we are content with obtaining a sample generated by a hybrid distribution $D(x) = (1-\delta)D_1(x) + \delta D_3(x)$ for some small enough $\delta \in (0,1)$. For this we can modify the above strategy which we now call `RejectionSample(m)`: Iterate through $m$ samples $x_1, \ldots, x_m$ from $D_2$, terminate if a sample $x_i$ is accepted with probability $\rho(x_i)$. If no sample is accepted, terminate with a sample from $D_3$.

It can be shown that the *failure* probability $\delta$ diminishes with increasing $m$. Indeed, we show that $\delta \leq e^{-m/\tau}$.

### 4.3. Application to `RS-k-means++`

Recall that our goal in `k-means++` is to sample the centers from the $D^2(\mathcal{X}, S)$ distribution, corresponding to $D_1$ in the previous discussion. We present two methods, one which samples the centers from the $D^2$ distribution and another which samples from a slightly 'perturbed' distribution $D_\delta^2(\mathcal{X}, S) = (1 - \delta)D^2(\mathcal{X}, S) + \delta \mathcal{U}[\mathcal{X}]$. This corresponds to $D_3(x) = \frac{1}{|\mathcal{X}|}$ simply being the uniform distribution over $\mathcal{X}$. We will use the `RejectionSample` and `RejectionSample(m)` for these methods respectively. In both cases we need to find a suitable distribution $D_2$ over $\mathcal{X}$ that $\tau$-oversamples $D_1$ (for a suitable $\tau$) and for which we obtain samples efficiently.

**Lemma 4.3.** *Let $S = \{c_1, \ldots, c_t\} \subset \mathcal{X}$ be chosen accord-*

*ing to the $D^2$ distribution (In particular, $c_1$ is a uniformly random point in $\mathcal{X}$). Let $D(\cdot; c_1)$ be the distribution over $\mathcal{X}$ defined by*

$$D(x; c_1) = \frac{\|x\|^2 + \|c_1\|^2}{\|X\|^2 + |\mathcal{X}|\|c_1\|^2}$$

*Then $D(x; c_1)$ $\tau$-oversamples $D^2(\mathcal{X}, S)$ for $\tau = 2\frac{\|X\|^2 + |\mathcal{X}|\|c_1\|^2}{\Delta(\mathcal{X}, S)}$.*

*Proof.* The choice of this distribution is given by the simple observation that $\Delta(x, S) = \min_{c \in S} \|x - c\|^2 \leq \|x - c_1\|^2 \leq 2(\|x\|^2 + \|c_1\|^2)$ where the final step uses the Cauchy-Schwarz inequality. Multiplying both sides by $\frac{1}{\Delta(\mathcal{X}, S)}$ gives the required result. □

How do we actually obtain samples from $D_2$? We can employ a simple technique: With probability $\frac{\|\mathcal{X}\|^2}{\|\mathcal{X}\|^2 + |\mathcal{X}|\|c_1\|^2}$, sample from $D_\mathcal{X}$ and with remaining probability, sample from $\mathcal{U}[\mathcal{X}]$. A standard calculation shows that this indeed gives the required distribution.

**Lemma 4.4.** *There is an algorithm that produces a sample from $\mathcal{X}$ according to the distribution $D(x; c_1)$ as defined in Lemma 4.3. Moreover it takes $O(\log |\mathcal{X}|)$ time after a one time preprocessing time of $\tilde{O}(\text{nnz}(\mathcal{X}))$.*

It can be seen that for these distributions, `RejectionSample` is equivalent to Procedure 3 with $m = \infty$. We then obtain the following:

**Lemma 4.5.** *Let $\epsilon \in (0,1)$ and $\mathcal{X} \subset \mathbb{R}^d$ be any dataset of $n$ points and $S = \{c_1, \ldots, c_t\} \subset \mathcal{X}$ such that $c_1$ is a uniformly random point in $\mathcal{X}$. Then Procedure 3 outputs a sample $c \in \mathcal{X}$ according to the distribution $D^2(\mathcal{X}, S)$. Moreover, the computational cost of the algorithm is bounded by $O(\beta(\mathcal{X}) \cdot (td + \log |\mathcal{X}|) \cdot \log(1/\epsilon))$ with probability atleast $1 - \epsilon$. Here, $\beta(\mathcal{X})$ is a parameter such that $\mathbb{E}[\beta(\mathcal{X})] \leq \frac{\Delta_1(\mathcal{X})}{\Delta_k(\mathcal{X})}$.*

We can apply this $D^2$ sampling technique $k$ times to obtain the centers according to `k-means++` . This is what `RS-k-means++`$(\mathcal{X}, k, \infty)$ does and it can be seen that Theorem 2.2 follows from Lemma 4.5; in particular the $8(\log k + 2)$ approximation guarantee of the sampled centers follows from the `k-means++` approximation guarantee by (Arthur & Vassilvitskii, 2007).

In our second approach, we replace `RejectionSample` by `RejectionSample(m)` which only repeats the rejection sampling loop $m$ times for a suitable choice of $m$. In particular, notice that Procedure 3 is essentially `RejectionSample(m)` for $D_1 = D^2(\mathcal{X}, S)$, $D_3$ being the uniform distribution over $\mathcal{X}$ and $D_2$ as in Lemma 4.3. This gives us the following result which is analogous to Lemma 4.5:

**Lemma 4.6.** *Let $m \in \mathbb{N}$ be a parameter, $\mathcal{X} \subset \mathbb{R}^d$ be any dataset of $n$ points and $S = \{c_1, \ldots, c_t\} \subset \mathcal{X}$ such that $c_1$ is a uniformly random point in $\mathcal{X}$. Then Procedure 3 with input $(\mathcal{X}, S, m \log t)$ outputs a sample $c \in \mathcal{X}$ according to the distribution $(1 - \delta) \cdot D^2(\mathcal{X}, S) + \delta \cdot \mathcal{U}[\mathcal{X}]$ for $\delta \leq e^{-m/\beta(\mathcal{X})}$. Moreover, the computational cost of the algorithm is bounded by $O(m \cdot (td + \log |\mathcal{X}|) \cdot \log t)$ with probability atleast $1 - \epsilon$. Here, $\beta(\mathcal{X})$ is a parameter such that $\mathbb{E}[\beta(\mathcal{X})] \leq \frac{\Delta_1(\mathcal{X})}{\Delta_k(\mathcal{X})}$.*

Again we can apply this sampling technique $k$ times to obtain $k$ centers and get towards Theorem 2.1. However, unlike Theorem 2.2, the approximation guarantee no longer follows directly from (Arthur & Vassilvitskii, 2007) because this sampling is from a 'perturbed' distribution from $D^2(\mathcal{X}, S)$. We consider this problem in the next section.

### 4.4. Analysis of $\delta$-$k$-`means++`

In order to analyze the solution quality of `RS-k-means++`, we examine an *abstract* variant of `k-means++` which we call $\delta$-`k-means++` . Instead of sampling from the $D^2$-distribution as in `k-means++` , we sample from a distribution which is a weighted average of the $D^2$-distribution and the uniform distribution with weights $(1 - \delta)$ and $\delta$ respectively. We show the following bound on the quality of solutions produced by this variant :

---

**Algorithm 4** $\delta$-`k-means++`$(\mathcal{X}, k, \delta)$

**Input :** dataset $\mathcal{X} \subset \mathbb{R}^d$, number of clusters $k \in \mathbb{N}$ and parameter $\delta \in (0, 1)$
**Output :** $S = \{c_1, \ldots, c_k\} \subset \mathcal{X}$
 1: Choose $c_1 \in \mathcal{X}$ uniformly at random and set $S \leftarrow \{c_1\}$
 2: **for** $t \in \{2, \ldots, k\}$ **do**
 3:   Chose a point $x \in \mathcal{X}$ with prob. $(1 - \delta)\frac{\Delta(x, S)}{\Delta(\mathcal{X}, S)} + \delta \frac{1}{|\mathcal{X}|}$
 4:   $c_t \leftarrow x, S \leftarrow S \cup \{c_t\}$
 5: **end for**
 6: **return** $S$

---

**Theorem 4.7.** *Let $\mathcal{X} \subset \mathbb{R}^d$ be any dataset which is to be partitioned into $k$ clusters. Let $S$ be the set of centers returned by $\delta$-`k-means++`$(\mathcal{X}, k, \delta)$ for any $\delta \in (0, 1)$. The following approximation guarantee holds :*

$$\mathbb{E}[\Delta(\mathcal{X}, S)] \leq 8(\ln k + 2)\Delta_k(\mathcal{X}) + \frac{6k\delta}{1 - \delta}\Delta_1(\mathcal{X})$$

From the previous discussions, Theorem 2.1 follows from Theorem 4.7 after substituting the expression for the failure probability $\delta$.

Our analysis uses the potential based approach introduced by (Dasgupta, 2013). Since we sample from a different distribution as compared to the standard `k-means++` , its

analysis does not directly carry over to $\delta$-`k-means++` . Indeed, it is known that the `k-means++` procedure is quite sensitive to even small changes in the $D^2$ distribution. This was first studied by (Bhattacharya et al., 2020) who were able to show only a $O(\log^2 k)$ guarantee when the centers are sampled from a distribution which is $\varepsilon$-close [13] to the exact $D^2$ distribution for a sufficiently small constant $\varepsilon$ . This result was recently improved upon by (Grunau et al., 2023) who recover the tight $O(\log k)$ guarantee of `k-means++` . In their analysis, (Grunau et al., 2023) incur a very large constant multiplicative blow-up [14] in the approximation guarantee and leave it as an open problem to show whether the true approximation guarantee can be bounded by a multiplicative factor of $1 + O(\varepsilon)$. In contrast, we show that the approximation guarantee of $\delta$-`k-means++` consists of an additive scale invariant variance factor proportional to $\delta$ in addition to the standard guarantee of `k-means++` with the same constants, which requires a careful analysis of propagation of the *extra* cost due to sampling from the uniform distribution in case a sample from the exact $D^2$ distribution is not generated by rejection sampling.

## 5. Experiments

In this section, we describe experimental validation of our theoretical results. The five data sets used for the experiments are described in Table 2. We also include an estimate of the data dependent parameter. This estimation was done by computing the ratio of the variance of the dataset with the clustering cost of the solution output by `RS-k-means++`$(\cdot, \cdot, \infty)$ averaged over 20 iterations.

**Observation 1.** The data dependent parameter $\beta$ does not take on prohibitively large values. Indeed, for the data sets used in our experiments, these values are quite reasonable. This observation is in accordance with the experiments of (Bachem et al., 2016b).

*Table 2.* Datasets used for experiments [16]

| $\mathcal{X}$ | $n$ | $k$ | $d$ | $\tilde{\beta}$ |
|---|---|---|---|---|
| DIABETES | $253,680$ | $50$ | $21$ | $\sim 6.5$ |
| FOREST | $581,010$ | $7$ | $54$ | $\sim 3.3$ |
| PROTEIN | $145,751$ | $100$ | $74$ | $\sim 9.7$ |
| POKER | $1,025,010$ | $50$ | $10$ | $\sim 2.4$ |
| CANCER | $94,730$ | $100$ | $117$ | $\sim 1.9$ |

Next, we compare the performance of the Monte Carlo

---

[13]Let $p(\cdot)$ and $p'(\cdot)$ represent probability mass functions over $\mathcal{X}$. $p'$ is said to be $\varepsilon$-close to $p$ if $|p'(x) - p(x)| \leq \varepsilon p(x) \, \forall \, x \in \mathcal{X}$.
[14]The constant blow-up of (Grunau et al., 2023) is bounded above by $\sum_{\ell=1}^{\infty} 90\ell e^{-\frac{\ell - 1}{40}} = \frac{90}{(1 - e^{-1/40})^2} \simeq 1.48 \times 10^5$

Markov Chain based algorithm AF-k-MC$^2$ of (Bachem et al., 2016a) using the prescribed chain length of $m = 200$ with our first algorithm i.e, RS-k-means++ without any upper-bound on the number of rejection-sampling rounds. We do so for each data set with various values of $k \in \{5, 10, 20, 50, 100\}$. The results are summarized in Table 3. The averages reported are over 20 iterations of each algorithm.

**Observation 2.** RS-k-means++ provides solutions with comparable quality to AF-k-MC$^2$, while generally being much faster. On datasets like POKER where the data size is much larger than the number of clusters, we observe a speedup of $\sim$ 40 - 70$\times$. Moreover, this version of RS-k-means++ does not require choosing any extra parameters as input.

*Table 3.* Comparison of AF-k-MC$^2(\cdot, \cdot, 200)$ with RS-k-means++$(\cdot, \cdot, \infty)$

| DATASET | $k$ | RS-k-means++ COST | AF-k-MC$^2$ COST | TIME RATIO |
|---------|-----|-------------------|------------------|------------|
| DIABETES | 5 | $2.847 \times 10^7$ | $3.089 \times 10^7$ | 37.5 |
| | 10 | $1.768 \times 10^7$ | $1.740 \times 10^7$ | 41.6 |
| | 20 | $1.174 \times 10^7$ | $1.195 \times 10^7$ | 31.4 |
| | 50 | $7.401 \times 10^6$ | $7.446 \times 10^6$ | 20.3 |
| | 100 | $5.515 \times 10^6$ | $5.476 \times 10^6$ | 13.5 |
| FOREST | 5 | $1.041 \times 10^{12}$ | $1.062 \times 10^{12}$ | 20.4 |
| | 10 | $5.941 \times 10^{11}$ | $5.853 \times 10^{11}$ | 21.9 |
| | 20 | $3.377 \times 10^{11}$ | $3.373 \times 10^{11}$ | 16.6 |
| | 50 | $1.834 \times 10^{11}$ | $1.846 \times 10^{11}$ | 9.1 |
| | 100 | $1.221 \times 10^{11}$ | $1.221 \times 10^{11}$ | 5.8 |
| PROTEIN | 5 | $1.048 \times 10^{12}$ | $1.085 \times 10^{12}$ | 25.0 |
| | 10 | $6.394 \times 10^{11}$ | $5.882 \times 10^{11}$ | 30.5 |
| | 20 | $4.388 \times 10^{11}$ | $4.434 \times 10^{11}$ | 23.2 |
| | 50 | $3.029 \times 10^{11}$ | $3.059 \times 10^{11}$ | 19.2 |
| | 100 | $2.417 \times 10^{11}$ | $2.456 \times 10^{11}$ | 11.9 |
| POKER | 5 | $7.81 \times 10^7$ | $8.03 \times 10^7$ | 71.5 |
| | 10 | $5.88 \times 10^7$ | $6.04 \times 10^7$ | 71.3 |
| | 20 | $4.58 \times 10^7$ | $4.51 \times 10^7$ | 60.7 |
| | 50 | $3.31 \times 10^7$ | $3.31 \times 10^7$ | 49.3 |
| | 100 | $2.68 \times 10^7$ | $2.69 \times 10^7$ | 40.1 |
| CANCER | 5 | $1.21 \times 10^7$ | $1.23 \times 10^7$ | 19.1 |
| | 10 | $1.07 \times 10^7$ | $1.06 \times 10^7$ | 34.1 |
| | 20 | $8.83 \times 10^6$ | $8.75 \times 10^6$ | 47.3 |
| | 50 | $7.02 \times 10^6$ | $7.06 \times 10^6$ | 54.0 |
| | 100 | $6.08 \times 10^6$ | $6.06 \times 10^6$ | 25.7 |

Next, we study the convergence properties of the version of RS-k-means++ which allows for a computational cost vs. solution quality trade-off. We plot the clustering cost of the solutions output against the time taken to generate them for various values of $m \in \{5, 10, 20, 50, 75, 100, 125, 150\}$. We compare the the solution quality with the baseline k-means++ solution as well.

**Observation 3.** The solution quality of RS-k-means++ approaches that of $k$-means ++ rapidly with increase in the upper bound for the number of rejection sampling rounds

allowed. This can be seen from the plots in Figure 5

More complete experimental details such as references for the data sets, variance for the clustering costs and data points for the plots can be found in Appendix C.

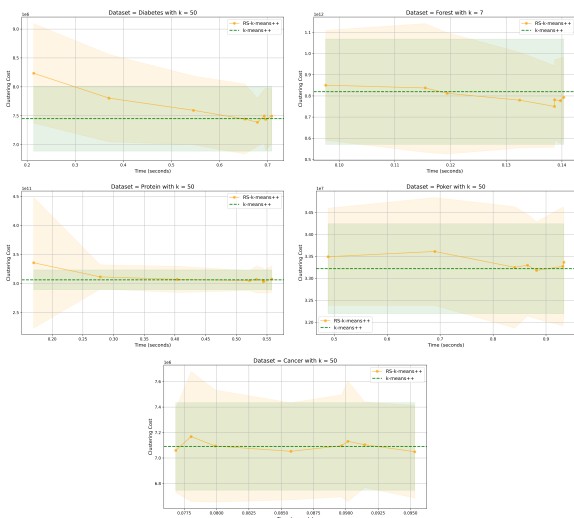

*Figure 1.* Trade-off plots showing the convergence of the solution quality of RS-k-means++ with respect to the $k$-means ++ baseline

## 6. Conclusion

In this work, we present a simple rejection sampling approach to $k$-means ++ through the RS-k-means++ algorithm. We show that our algorithm allows for new trade-offs between the computational cost and solution quality of the $k$-means ++ seeding procedure. It also has the advantage of supporting fast data updates and being easy to adapt in the parallel setting. The solution quality of RS-k-means++ is bounded through the analysis of a *perturbed* version of the standard $k$-means ++ method. The effectiveness of our approach is reflected in the experimental evaluations performed. Interesting future directions include the possibility of improving the dependence of the runtime - quality trade-off on the data dependent parameter. We believe that similar techniques could be adapted to the setting where the data set is present in the disk instead of the main memory and the goal is to minimize the number of disk accesses.

## Impact Statement

Our work focuses on speeding-up the $k$-means ++ algorithm for clustering. Therefore, we expect that our new algorithm could have impact on domains in which clustering plays an important role. A concrete impact in society is harder to predict since this is mainly fundamental research.

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

# Appendix

## A. Rejection Sampling

Given the dataset $\mathcal{X} \subset \mathbb{R}^d$ and a set of already chosen centers $S \subset \mathcal{X}$, our goal is to obtain a sample from $\mathcal{X}$ according to the $D^2(\mathcal{X}, S)$ distribution. Recall that we defined the distribution $D_{\mathcal{X}}$ over $\mathcal{X}$ by $D_{\mathcal{X}}(x) = \frac{\|x\|^2}{\|\mathcal{X}\|^2}$. The main ingredient of our algorithm is a rejection sampling procedure which allows us convert samples from $D_{\mathcal{X}}$ to a sample from $D^2(\mathcal{X}, S)$.

We shall pre-process our dataset so that we can efficiently sample from $D_{\mathcal{X}}$, and then convert samples from $D_{\mathcal{X}}$ to samples from $D^2(\mathcal{X}, S)$. Choosing the first center uniformly at random from $\mathcal{X}$ and repeating this procedure for $k - 1$ times is precisely our algorithm for performing the $k$-`means ++` seeding.

**Definition A.1.** Suppose $D_1$, $D_2$ define probability distributions over $\mathcal{X}$. The distribution $D_2$ is said to $\tau$-*oversample* $D_1$ for $\tau > 0$ if $D_1(x) \leq \tau D_2(x)$ for each $x \in \mathcal{X}$.

---

**Algorithm 5** `RejectionSample`

---

**Input:** Samples generated from $D_2$
**Output:** A sample generated from $D_1$

1: `sampled` $=$ `False`
2: **repeat**
3:    $x \sim D_2$ , $r \sim [0, 1]$
4:    Compute $\rho(x) = \frac{D_1(x)}{\tau D_2(x)}$
5:    **if** $r \leq \rho(x)$ **then**
6:       **output** $x$ and set `sampled` $=$ `True`
7:    **end if**
8: **until** `sampled` $=$ `True`

---

Consider Algorithm 5 which takes samples generated from $D_2$ as input and outputs a sample generated from $D_1$.

**Lemma A.2.** *Let $D_1, D_2$ be probability distributions over $\mathcal{X}$ such that $D_2$ $\tau$-oversamples $D_1$. The expected number of samples from $D_2$ required by* `RejectionSample` *to output a single sample from $D_1$ is $\tau$. Moreover, for any $\varepsilon \in (0, 1)$ the probability that more than $\tau \ln \frac{1}{\varepsilon}$ samples are required is atmost $\varepsilon$.*

*Proof.* Let $T$ be the random variable denoting the number of rounds required for a sample to be output by `RejectionSample`. Let `Output` denote the event that a sample is output in a particular round. We have

$$\Pr[\texttt{Output}] = \sum_{x \in \mathcal{X}} D_2(x)\rho(x) = \tau^{-1} \sum_{x \in \mathcal{X}} D_1(x) = \tau^{-1}$$

Given that a sample is generated, it is easy to see that it is distributed according to $D_1$. It takes exactly $t$ rounds for a sample to be generated if no sample is generated in the first $t - 1$ iterations and a sample is generated in the last iteration. Hence,

$$\Pr[T = t] = (\Pr[\neg\texttt{Output}])^{t-1} \Pr[\texttt{Output}] = \tau^{-1}(1 - \tau^{-1})^{t-1}$$

which means that $T$ is a geometric random variable with parameter $\tau^{-1}$, so that $E[T] = \tau$. It is easy to see that $T$ has exponentially diminishing tails :

$$\Pr[T > t] = \sum_{j=t+1}^{\infty} \tau^{-1}(1 - \tau^{-1})^{j-1} = (1 - \tau^{-1})^t \leq e^{-t/\tau}$$

from which the lemma follows. $\qquad\square$

*Remark* A.3. Note that the algorithm does not require any estimate on the value of $\tau$, computing which may be non-trivial. It only requires the ability to compute the ratio $\rho(x) = \frac{D_1(x)}{\tau D_2(x)}$ for each $x \in \mathcal{X}$.

In the current form, Algorithm 5 does not have any control over the number of samples from $D_2$ which it may need to examine. However, a bound on the number of samples to be examined can be used if we are content with sampling from a slightly perturbed distribution. Suppose we have another distribution $D_3$ over $\mathcal{X}$. This time we are allowed to use samples coming from $D_2$ and $D_3$ and instead of a sample from $D_1$, we are content with obtaining a sample generated by a hybrid distribution $D(x) = (1 - \delta)D_1(x) + \delta D_3(x)$ for some small enough $\delta \in (0, 1)$. For this we can modify Algorithm 5 to Algorithm 6 as follows :

---

**Algorithm 6** RejectionSample($m$)

---

**Input:** Samples generated from $D_2, D_3$
**Output:** A sample $x$ with probability $D(x) = (1 - \delta)D_1(x) + \delta D_3(x)$ where $\delta \leq e^{-m/\tau}$

1: `sampled = False` , `iter` $= 0$
2: **repeat**
3:   `iter = iter` $+ 1$
4:   $x \sim D_2$ , $r \sim [0, 1]$
5:   Compute $\rho(x) = \frac{D_1(x)}{\tau D_2(x)}$
6:   **if** $r \leq \rho(x)$ **then**
7:     **output** $x$ and set `sampled = True`
8:   **end if**
9: **until** `sampled = True` **or** `iter` $> m$
10: **if** `sampled = False` **then**
11:   **output** $x \sim D_3$
12: **end if**

---

**Lemma A.4.** *Let $m > 0$ be the upper bound on the number of rounds in* RejectionSample. *The output samples come from a distribution $D$ which can be expressed as $D(x) = (1 - \delta)D_1(x) + \delta D_3(x)$ where $\delta \leq e^{-m/\tau}$.*

*Proof.* A point is sampled from $D_3$ if and only if no sample is generated in $m$ rounds of rejection sampling. Hence, the probability of sampling a point $x \in \mathcal{X}$ is :

$$\Pr[x \sim \texttt{RejectionSample}(m)]$$
$$= \Pr[x|T \leq m]\Pr[T \leq m] + \Pr[x|T > m]\Pr[T > m]$$
$$= (1 - \delta)D_1(x) + \delta D_3(x)$$

where $\delta = \Pr[T > m] \leq e^{-m/\tau}$ from the proof of Lemma A.2. $\square$

### A.1. Application to `RS-k-means++`

Recall that our goal in `k-means++` is to sample the centers from the distribution $D^2(\mathcal{X}, S)$ over $\mathcal{X}$ given by $D^2(\mathcal{X}, S) = D_1(x) = \frac{\Delta(x,S)}{\Delta(\mathcal{X},S)}$. In this work we present two methods, one which samples the centers from the $D^2$ distribution and another which samples from a slightly 'perturbed' distribution $D(x) = (1 - \delta)D_1(x) + \delta D_3(x)$ where $D_3(x) = \frac{1}{|\mathcal{X}|}$ is simply the uniform distribution over $\mathcal{X}$. We will use the RejectionSample and RejectionSample($m$) for these methods respectively. In both cases we need to find a suitable distribution $D_2$ over $\mathcal{X}$ that $\tau$-*oversamples* $D_1$ (for a suitable $\tau$) and for which we have an efficient method to obtain samples.

**Lemma A.5.** *Let $S = \{c_1, \ldots, c_t\} \subset \mathcal{X}$ be chosen according to the $D^2$ distribution (In particular, $c_1$ is a uniformly random point in $\mathcal{X}$). Let $D_2$ be the distribution over $\mathcal{X}$ defined by*

$$D_2(x) = \frac{\|x\|^2 + \|c_1\|^2}{\|X\|^2 + |\mathcal{X}|\|c_1\|^2}$$

*Then $D_2$ $\tau$-oversamples $D^2(\mathcal{X}, S)$ for $\tau = 2\frac{\|X\|^2 + |\mathcal{X}|\|c_1\|^2}{\Delta(\mathcal{X},S)}$.*

*Proof.*

$$\Delta(x, S) = \min_{c \in S} \|x - c\|^2 \le \|x - c_1\|^2 \le 2(\|x\|^2 + \|c_1\|^2)$$

where the final inequality is obtained via the Cauchy-Schwarz inequality. Multiplying both sides by $\frac{1}{\Delta(\mathcal{X},S)}$ gives the required result. □

An immediate issue is: how do we actually obtain samples from $D_2$? We will deal with this issue in a bit; for now, assume that we can efficiently obtain such samples after a preprocessing step.

With this, we can apply Algorithm 5 for $D_1$ being the required $D^2(\mathcal{X}, S)$ distribution and $D_2$ as in Lemma A.5. It can be seen that for these distributions, Algorithm 5 is equivalent to Procedure 3 where $m = \infty$. Thus Lemma A.2 gives the following Corollary.

**Corollary A.6.** *Let $\epsilon \in (0, 1)$ and $\mathcal{X} \subset \mathbb{R}^d$ be any dataset of $n$ points and $S = \{c_1, \dots, c_t\} \subset \mathcal{X}$ such that $c_1$ is a uniformly random point in $\mathcal{X}$. Assume that we can obtain a sample from the following distribution over $\mathcal{X}$ in $O(\log |\mathcal{X}|)$ time:*

$$D_2(x) = \frac{\|x\|^2 + \|c_1\|^2}{\|X\|^2 + |\mathcal{X}|\|c_1\|^2}$$

*Then Procedure 3 outputs a sample $c \in \mathcal{X}$ according to the distribution $D^2(\mathcal{X}, S)$. Moreover, the computational cost of the algorithm is bounded by $O(\beta(\mathcal{X}) \cdot (td + \log |\mathcal{X}|) \cdot \log(1/\epsilon))$ with probability atleast $1 - \epsilon$. Here, $\beta(\mathcal{X})$ is a parameter such that $\mathbb{E}[\beta(\mathcal{X})] \le \frac{\Delta_1(\mathcal{X})}{\Delta_k(\mathcal{X})}$.*

*Proof.* By Lemma A.5 we know that $D_2$ $\tau$-oversamples $D^2(\mathcal{X}, S)$ for $\tau = 2\frac{\|X\|^2 + |\mathcal{X}|\|c_1\|^2}{\Delta(\mathcal{X},S)} \le 2\frac{\|X\|^2 + |\mathcal{X}|\|c_1\|^2}{\Delta_k(\mathcal{X})}$. So

$$\mathbb{E}[\tau] \le 2\frac{\|X\|^2 + |\mathcal{X}|\mathbb{E}[\|c_1\|^2]}{\Delta_k(\mathcal{X})}$$

$$= 2\frac{\|X\|^2 + |\mathcal{X}| \cdot \frac{1}{|\mathcal{X}|}\sum_{x \in \mathcal{X}}\|x\|^2}{\Delta_k(\mathcal{X})}$$

$$= \frac{4\|\mathcal{X}\|^2}{\Delta_k(\mathcal{X})} = \frac{4\Delta_1(\mathcal{X})}{\Delta_k(\mathcal{X})}$$

where the last inequality follows from the fact that $\mathcal{X}$ is translated so that its centroid is at the origin.

Thus applying Lemma A.2 gives us the result for $\beta(\mathcal{X}) = \tau/4$. □

We can apply this $D^2$ sampling technique $k$ times to obtain the centers according to `k-means++` . This is what `RS-k-means++`$(\mathcal{X}, k, \infty)$ does and it can be seen that Theorem 2.2 follows from Corollary A.6; in particular the approximation guarantee of the sampled centers follows from (Arthur & Vassilvitskii, 2007).

In our second approach, we replace `RejectionSample` by `RejectionSample`$(m)$ which only repeats the rejection sampling loop $m$ times for a suitable choice of $m$. In particular, notice that Procedure 3 is essentially Algorithm 6 for $D_1 = D^2(\mathcal{X}, S)$, $D_3$ being the uniform distribution over $\mathcal{X}$ and $D_2$ as in Lemma A.5.

This gives us the following corollary whose proof can be argued analogous to A.6:

**Corollary A.7.** *Let $m \in \mathbb{N}$ be a parameter, $\mathcal{X} \subset \mathbb{R}^d$ be any dataset of $n$ points and $S = \{c_1, \dots, c_t\} \subset \mathcal{X}$ such that $c_1$ is a uniformly random point in $\mathcal{X}$. Assume that we can obtain a sample from the following distribution over $\mathcal{X}$ in $O(\log |\mathcal{X}|)$ time:*

$$D_2(x) = \frac{\|x\|^2 + \|c_1\|^2}{\|X\|^2 + |\mathcal{X}|\|c_1\|^2}$$

*Then Procedure 3 with input $(\mathcal{X}, S, m \log t)$ outputs a sample $c \in \mathcal{X}$ according to the distribution $(1 - \delta) \cdot D^2(\mathcal{X}, S) + \delta \cdot \mathcal{U}[\mathcal{X}]$ for $\delta \le e^{-m/\beta(\mathcal{X})}$. Moreover, the computational cost of the algorithm is bounded by $O(m \cdot (td + \log |\mathcal{X}|) \cdot \log t)$ with probability at least $1 - \epsilon$. Here, $\beta(\mathcal{X})$ is a parameter such that $\mathbb{E}[\beta(\mathcal{X})] \le \frac{\Delta_1(\mathcal{X})}{\Delta_k(\mathcal{X})}$.*

Again we can apply this sampling technique $k$ times to obtain $k$ centers. Note that this sampling is from a 'perturbed' distribution from $D^2(\mathcal{X}, S)$, so the approximation guarantee no longer follows directly from (Arthur & Vassilvitskii, 2007). However we analyse this in Appendix B to get the following:

**Theorem A.8.** *Let $\mathcal{X} \subset \mathbb{R}^d$ be any dataset which is to be partitioned into $k$ clusters. Let $S$ be the set of centers returned by $\delta$-`k-means++`$(\mathcal{X}, k, \delta)$ for any $\delta \in (0, 0.5)$. The following approximation guarantee holds :*

$$\mathbb{E}[\Delta(\mathcal{X}, S)] \leq 8(\ln k + 2)\Delta_k(\mathcal{X}) + \frac{6k\delta}{1-\delta}\Delta_1(\mathcal{X})$$

which will finally prove Theorem 2.1 after substituting value of the *failure probability $\delta$*.

In the following section we show how to obtain samples from $D_2$.

### A.2. Sampling from $D_2$ via a Preprocessed Data Structure

Given $\mathcal{X} \subset \mathbb{R}^d$, consider the vector $v_{\mathcal{X}} \in \mathbb{R}^{|\mathcal{X}|}$ given by $v_{\mathcal{X}}(x) = \|x\|$. Define $D_{\mathcal{X}}(x) = \frac{\|x\|^2}{\|\mathcal{X}\|^2}$ as a distribution over $\mathcal{X}$. We will use a (complete) binary tree data structure to sample from $D_{\mathcal{X}}$. The leaves of the binary tree correspond to the entries of $v_{\mathcal{X}}$ and store weight $v_{\mathcal{X}}(x)^2$ along with the sign of $v_{\mathcal{X}}(x)$. The internal nodes also store a weight that is equal to the sum of weights of its children. To sample from $D_{\mathcal{X}}$, we traverse the tree, choosing either to go left or right at each node with probability proportional to the weight of its two children until reaching the leaves. The binary tree similarly supports querying and updating the entries of $v_{\mathcal{X}}$.

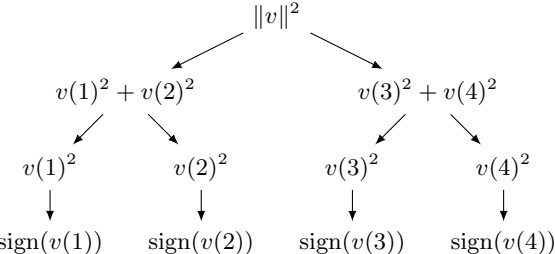

Figure 2. Data structure for sampling from a vector $v \in \mathbb{R}^4$

We state this formally following (Tang, 2019), in which such data structures, called *sample and query access data structures* were introduced.

**Lemma A.9.** *(Lemma 3.1 in (Tang, 2019)) There exists a data structure storing a vector $v \in \mathbb{R}^n$ with $\nu$ nonzero entries in $O(\nu \log n)$ space which supports the following operations:*

- *Reading and updating an entry of $v$ in $O(\log n)$ time*

- *Finding $\|v\|^2$ in $O(1)$ time*

- *Generating an independent sample $i \in \{1, \ldots, n\}$ with probability $\frac{v(i)^2}{\sum_{j=1}^n v(j)^2}$ in $O(\log n)$ time.*

Note that if $n$ is not a perfect power of 2 then we can find a $n' \in \mathbb{N}$ which is a perfect power of 2 such that $n' < n < 2n'$. We can then set the remaining $2n' - n$ data points to have zero norm and use this dataset instead to construct the complete binary tree. Thus the following corollary is immediate.

**Corollary A.10.** *There is a data structure that can be prepared in $\tilde{O}(\texttt{nnz}(\mathcal{X}))$ time which enables generating a sample from $D_{\mathcal{X}}$ in $O(\log |\mathcal{X}|)$ time.*

We will now show how to sample from $D_2$ in $O(\log |\mathcal{X}|)$ time by Procedure 7.

**Lemma A.11.** *Procedure 7 produces a sample from $\mathcal{X}$ according to the distribution $D_2$ as defined in Lemma A.5. Moreover it takes $O(\log |\mathcal{X}|)$ time after a one time preprocessing time of $\tilde{O}(\texttt{nnz}(\mathcal{X}))$.*

---
**Procedure 7** `SampleDistribution`

---
**Input:** A center $c \in \mathcal{X}$

**Output:** A sample according to the distribution $D_2$ defined as $D_2(x) = \frac{\|x\|^2 + \|c\|^2}{\|X\|^2 + |\mathcal{X}|\|c\|^2}$

  1: Generate $r \sim \mathcal{U}[0, 1]$

  2: **if** $r \leq \frac{\|\mathcal{X}\|^2}{\|\mathcal{X}\|^2 + |\mathcal{X}|\|c\|^2}$ **then**

  3:    Generate a sample $x \sim D_{\mathcal{X}}$ using the data structure from Section A.2

  4: **else**

  5:    Generate a sample $x \sim \mathcal{U}[\mathcal{X}]$

  6: **end if**

  7: **output** $x$

---

*Proof.* The probability of a sampled point is as follows:

$$\Pr[x] = \Pr\left[r \leq \frac{\|\mathcal{X}\|^2}{\|\mathcal{X}\|^2 + n\|c_1\|^2}\right] \Pr[x \sim D_{\mathcal{X}}]$$

$$+ \Pr\left[r > \frac{\|\mathcal{X}\|^2}{\|\mathcal{X}\|^2 + |\mathcal{X}|\|c_1\|^2}\right] \Pr[x \sim \mathcal{U}[\mathcal{X}]]$$

$$= \frac{\|\mathcal{X}\|^2}{\|V\|^2 + |\mathcal{X}|\|c_1\|^2} \frac{\|x\|^2}{\|\mathcal{X}\|^2} + \frac{|\mathcal{X}|\|c_1\|^2}{\|\mathcal{X}\|^2 + |\mathcal{X}|\|c_1\|^2} \frac{1}{|\mathcal{X}|}$$

$$= \frac{2(\|v_i\|^2 + \|c_1\|^2)}{2(\|\mathcal{X}\|^2 + |\mathcal{X}|\|c_1\|^2)} = D_2(x)$$

The time complexity follows from A.10. $\qquad\qquad\square$

## B. Analysis of $\delta$-$k$-`means++`

In order to analyze the solution quality of `RS-k-means++`, we examine an *abstract* variant of `k-means++` which we call $\delta$-`k-means++` . Instead of sampling from the $D^2$-distribution as in `k-means++` , we sample from a distribution which is a weighted average of the $D^2$-distribution and the uniform distribution with weights $(1 - \delta)$ and $\delta$ respectively. We refer to this distribution on $\mathcal{X}$ as $D_\delta^2(\mathcal{X}, S)$ for some set of centers $S \subset \mathcal{X}$ . When clear from context, we just use $D_\delta^2$.

---
**Algorithm 8** $\delta$-k-means++$(\mathcal{X}, k, \delta)$

---
**Input :** dataset $\mathcal{X} \subset \mathbb{R}^d$, number of clusters $k \in \mathbb{N}$ and parameter $\delta \in (0, 1)$

**Output :** $S = \{c_1, \ldots, c_k\} \subset \mathcal{X}$

  1: Choose $c_1 \in \mathcal{X}$ uniformly at random and set $S \leftarrow \{c_1\}$

  2: **for** $t \in \{2, \ldots, k\}$ **do**

  3:    Chose a point $x \in \mathcal{X}$ with prob. $(1 - \delta)\frac{\Delta(x,S)}{\Delta(\mathcal{X},S)} + \delta\frac{1}{|\mathcal{X}|}$

  4:    $c_t \leftarrow x$

  5:    $S \leftarrow S \cup \{c_t\}$

  6: **end for**

  7: **return** $S$

---

The main objective of this section is to prove the following :

**Theorem B.1.** *Let $\mathcal{X} \subset \mathbb{R}^d$ be any dataset which is to be partitioned into $k$ clusters. Let $S$ be the set of centers returned by $\delta$-`k-means++`$(\mathcal{X}, k, \delta)$ for any $\delta \in (0, 0.5)$ . The following approximation guarantee holds :*

$$\mathbb{E}[\Delta(\mathcal{X}, S)] \leq 8(\ln k + 2)\Delta_k(\mathcal{X}) + \frac{6k\delta}{1 - \delta}\Delta_1(\mathcal{X})$$

Our analysis closely follows the potential based approach of (Dasgupta, 2013). Since we sample from a different distribution as compared to the standard `k-means++` , its analysis does not directly carry over to $\delta$-`k-means++` . Indeed, it is known

that the k-means++ procedure is quite sensitive to even small changes in the $D^2$ distribution. This was first studied by (Bhattacharya et al., 2020) who were able to show only a $O(\log^2 k)$ guarantee when the centers are sampled from a distribution which is $\varepsilon$-close [17] to the exact $D^2$ distribution for a sufficiently small constant $\varepsilon$. This result was recently improved upon by (Grunau et al., 2023) who recover the tight $O(\log k)$ guarantee of k-means++ . In their analysis, (Grunau et al., 2023) incur a very large constant multiplicative blow-up [18] in the approximation guarantee and leave it as an open problem to show whether the true approximation guarantee can be bounded by a multiplicative factor of $1 + O(\varepsilon)$. In contrast, we show that the approximation guarantee of $\delta$-k-means++ consists of an additive scale invariant variance factor proportional to $\delta$ in addition to the standard guarantee of k-means++ with the same constants.

### B.1. Some Useful Lemmas

In this section, we state some crucial lemmas that shall be helpful in the analysis. Throughout our work, the centroid of a set of points $\mathcal{P} \subset \mathbb{R}^d$ is denoted by $\mu(\mathcal{P}) = \frac{1}{|\mathcal{P}|} \sum_{p \in \mathcal{P}} p$.

The following folklore lemma is analogous to the bias-variance decomposition in machine learning.

**Lemma B.2.** *For any set of points $\mathcal{P} \subset \mathbb{R}^d$ and any point $z \in \mathbb{R}^d$ (possibly not in $\mathcal{P}$), the following holds :*

$$\Delta(\mathcal{P}, z) = \Delta(\mathcal{P}, \mu(\mathcal{P})) + |\mathcal{P}| \|z - \mu(\mathcal{P})\|^2$$

This shows that the solution for the 1-means problem is the centroid of the data set i.e, $\Delta_1(\mathcal{P}) = \Delta(\mathcal{P}, \mu(\mathcal{P}))$. The above lemma can be easily used to show the following.

**Lemma B.3.** *(Lemma 3.1 in (Arthur & Vassilvitskii, 2007)) For any set of points $\mathcal{P} \subset \mathbb{R}^d$, if a point $z \in \mathcal{P}$ is chosen uniformly at random, then the following holds :*

$$\mathbb{E}[\Delta(\mathcal{P}, z)] = 2\Delta(\mathcal{P}, \mu(\mathcal{P}))$$

We now state some useful bounds on the probability that a point is chosen from the $\mathcal{D}_\delta^2$ distribution with respect to some centers $S$ conditioned on it coming from a particular subset of points. For a point $z \in \mathbb{R}^d$ (possibly chosen randomly from some probability distribution) and a set of points $\mathcal{P} \subset \mathbb{R}^d$, we denote the event $\{z \in \mathcal{P}\}$ by $\chi_{\mathcal{P}}(z)$.

**Lemma B.4.** *Let $\mathcal{P} \subset \mathbb{R}^d$ be a set of points with $\mathcal{Q} \subset \mathcal{P}$ being an arbitrary subset of $\mathcal{P}$ such that $|\mathcal{Q}| \neq 0$. Let $S \subset \mathcal{P}$ be a set of cluster centers. For any point $z \in \mathcal{Q}$ and parameter $\delta \in (0,1)$, the following hold :*

1. $\Pr[z \sim \mathcal{D}_\delta^2 | \chi_{\mathcal{Q}}(z)] \leq \frac{\Delta(z)}{\Delta(\mathcal{Q})} + \frac{\delta}{1-\delta} \frac{1}{|\mathcal{P}|} \frac{\Delta(\mathcal{P})}{\Delta(\mathcal{Q})}$

2. $\Pr[z \sim \mathcal{D}_\delta^2 | \chi_{\mathcal{Q}}(z)] \geq \frac{\Delta(z)}{\Delta(\mathcal{Q})} - \frac{\delta}{1-\delta} \frac{|\mathcal{Q}|}{|\mathcal{P}|} \frac{\Delta(z)\Delta(\mathcal{P})}{\Delta(\mathcal{Q})^2}$

*Here, $\Delta(\cdot)$ denotes $\Delta(\cdot, S)$ for simplicity.*

*Proof.* The probability that a chosen point belongs to $\mathcal{Q}$ is

$$\Pr[z \sim \mathcal{D}_\delta^2 \cap \chi_{\mathcal{Q}}(z)] = \sum_{q \in \mathcal{Q}} \left( (1-\delta) \frac{\Delta(q)}{\Delta(\mathcal{P})} + \delta \frac{1}{|\mathcal{P}|} \right)$$

$$= (1-\delta) \frac{\Delta(\mathcal{Q})}{\Delta(\mathcal{P})} + \delta \frac{|\mathcal{Q}|}{|\mathcal{P}|}$$

Hence the required conditional probability is

$$\Pr[z \sim \mathcal{D}_\delta^2 | \chi_{\mathcal{Q}}(z)] = \frac{(1-\delta) \frac{\Delta(z)}{\Delta(\mathcal{P})} + \delta \frac{1}{|\mathcal{P}|}}{(1-\delta) \frac{\Delta(\mathcal{Q})}{\Delta(\mathcal{P})} + \delta \frac{|\mathcal{Q}|}{|\mathcal{P}|}}$$

---

[17] Let $p(\cdot)$ and $p'(\cdot)$ represent probability mass functions over $\mathcal{X}$. $p'$ is said to be $\varepsilon$-close to $p$ if $|p'(x) - p(x)| \leq \varepsilon p(x) \, \forall x \in \mathcal{X}$.

[18] The constant blow-up of (Grunau et al., 2023) is bounded above by $\sum_{\ell=1}^{\infty} 90\ell e^{-\frac{\ell-1}{40}} = \frac{90}{(1-e^{-1/40})^2} \simeq 1.48 \times 10^5$

For 1. we have :

$$\frac{(1-\delta)\frac{\Delta(z)}{\Delta(\mathcal{P})}+\delta\frac{1}{|\mathcal{P}|}}{(1-\delta)\frac{\Delta(\mathcal{Q})}{\Delta(\mathcal{P})}+\delta\frac{|\mathcal{Q}|}{|\mathcal{P}|}} \leq \frac{(1-\delta)\frac{\Delta(z)}{\Delta(\mathcal{P})}+\delta\frac{1}{|\mathcal{P}|}}{(1-\delta)\frac{\Delta(\mathcal{Q})}{\Delta(\mathcal{P})}}$$

$$= \frac{\Delta(z)}{\Delta(\mathcal{Q})} + \frac{\delta}{1-\delta}\frac{1}{|\mathcal{P}|}\frac{\Delta(\mathcal{P})}{\Delta(\mathcal{Q})}$$

For 2. we have :

$$\frac{(1-\delta)\frac{\Delta(z)}{\Delta(\mathcal{P})}+\delta\frac{1}{|\mathcal{P}|}}{(1-\delta)\frac{\Delta(\mathcal{Q})}{\Delta(\mathcal{P})}+\delta\frac{|\mathcal{Q}|}{|\mathcal{P}|}} = \frac{\Delta(z)}{\Delta(\mathcal{Q})}\left(\frac{1+\frac{\delta}{1-\delta}\frac{\Delta(\mathcal{P})}{|\mathcal{P}|\Delta(z)}}{1+\frac{\delta}{1-\delta}\frac{|\mathcal{Q}|\Delta(\mathcal{P})}{|\mathcal{P}|\Delta(Q)}}\right)$$

$$\geq \frac{\Delta(z)}{\Delta(\mathcal{Q})}\left(1+\frac{\delta}{1-\delta}\frac{|\mathcal{Q}|\Delta(\mathcal{P})}{|\mathcal{P}|\Delta(Q)}\right)^{-1}$$

$$\geq \frac{\Delta(z)}{\Delta(\mathcal{Q})}\left(1-\frac{\delta}{1-\delta}\frac{|\mathcal{Q}|\Delta(\mathcal{P})}{|\mathcal{P}|\Delta(Q)}\right)$$

where in the last step we used the fact that $(1+x)^{-1} \geq 1-x$ for any $x \geq 0$. This completes the proof of the lemma. $\square$

Recall that $\mathrm{OPT}_k = \{c_1^*, \ldots c_k^*\}$ represented the optimal set of centers of the k-means problem for the dataset $\mathcal{X}$. For a center $c_i \in \mathrm{OPT}_k$, let us denote the set of all points in $\mathcal{X}$ closer to $c_i$ than any other center in $\mathrm{OPT}_k$ by $\mathcal{C}_i$. Note $c_i^* = \mu(\mathcal{C}_i)$ from Lemma B.2.

Next, we show a bound on the expected cost of a cluster $\mathcal{C}_i$ of $\mathrm{OPT}_k$ when a point is added to the current set of clusters from $\mathcal{C}_i$ through the $D_\delta^2$ distribution. The following is analogous to Lemma 3.2 of (Arthur & Vassilvitskii, 2007) with an additional factor depending on $\delta$.

**Lemma B.5.** *Let $\mathcal{X} \subset \mathbb{R}^d$ be any dataset. Suppose we have a set of already chosen cluster centers $S$ and a new center $z$ is added to $S$ from the set of points $\mathcal{C}_i$ in the cluster corresponding to some $c_i \in \mathrm{OPT}_k$ through the $D_\delta^2(\mathcal{X}, S)$ distribution. The following holds :*

$$\mathbb{E}[\Delta(\mathcal{C}_i, S \cup \{z\})|S, \chi_{\mathcal{C}_i}(z)] \leq 8\Delta(\mathcal{C}_i, \mu(\mathcal{C}_i)) + \frac{\delta}{1-\delta}\frac{|\mathcal{C}_i|}{|\mathcal{X}|}\Delta(\mathcal{X}, S)$$

*Proof.* When the new center $z$ is added, each point $x \in \mathcal{C}$ contributes

$$\Delta(x, S \cup \{z\}) = \min(\|x-z\|^2, \Delta(x, S))$$

to the overall cost. The expected cost of the cluster $\mathcal{C}$ can hence be written as :

$$\mathbb{E}[\Delta(\mathcal{C}_i, S \cup \{z\})|S, \chi_{\mathcal{C}_i}(z)]$$

$$= \sum_{z \in \mathcal{C}_i} \Pr[z \sim \mathcal{D}_\delta^2|S, \chi_{\mathcal{C}_i}(z)] \sum_{x \in \mathcal{C}_i} \Delta(x, S \cup \{z\})$$

$$\leq \sum_{z \in \mathcal{C}_i} \frac{\Delta(z, S)}{\Delta(\mathcal{C}_i, S)} \sum_{x \in \mathcal{C}_i} \min(\|x-z\|^2, \Delta(x, S))$$

$$+ \sum_{z \in \mathcal{C}_i} \left(\frac{\delta}{1-\delta}\frac{1}{|\mathcal{X}|}\frac{\Delta(\mathcal{X}, S)}{\Delta(\mathcal{C}_i, S)}\right) \sum_{x \in \mathcal{C}_i} \min(\|x-z\|^2, \Delta(x, S))$$

where in the last step we used item (i) of Lemma B.4. From Lemma 3.2 of (Arthur & Vassilvitskii, 2007), the first term is bounded above by $8\Delta(\mathcal{C}_i, \mu(\mathcal{C}_i))$. Let us focus on the second term. Noting that

$$\sum_{x \in \mathcal{C}_i} \min(\|x-z\|^2, \Delta(x, S)) \leq \sum_{x \in \mathcal{C}_i} \Delta(x, S) = \Delta(\mathcal{C}_i, S)$$

the second term can be bounded above by the following :

$$\sum_{z \in \mathcal{C}_i} \left( \frac{\delta}{1-\delta} \frac{1}{|\mathcal{X}|} \frac{\Delta(\mathcal{X}, S)}{\Delta(\mathcal{C}_i, S)} \right) \Delta(\mathcal{C}_i, S) = \frac{\delta}{1-\delta} \frac{|\mathcal{C}_i|}{|\mathcal{X}|} \Delta(\mathcal{X}, S)$$

from which the lemma follows. $\qquad \square$

### B.2. Main Analysis

Before getting into the proof, let us set up some notation.

**Notation.** Let $t \in \{1, \ldots. k\}$ denote the number of centers already chosen by $\delta$-`k-means++` . Let $S_t := \{c_1, \ldots, c_t\}$ be the set of centers after iteration $t$ . We say that a cluster $\mathcal{C}_i$ of $\text{OPT}_k$ is *covered* by $S_t$ if at least one of its centers is in $\mathcal{C}_i$. If not, then it is *uncovered*. We denote

$$H_t = \{i \in \{1, \ldots, k\} : \mathcal{C}_i \cap S_t \neq \emptyset\}, \ U_t = \{1, \ldots, k\} \backslash H_t$$

Similarly, the dataset $\mathcal{X}$ can be partitioned in two parts : $\mathcal{H}_t \subset \mathcal{X}$ being the points belonging to *covered* clusters and $\mathcal{U}_t = \mathcal{X} \backslash \mathcal{H}_t$ being the points belonging to *uncovered* clusters. Let $W_t = t - |H_t|$ denote the number of *wasted* iterations so far i.e, the number of iterations in which no new cluster was covered. Note that we always have $|H_t| \leq t$ and hence $|U_t| \geq k - t$. For any $\mathcal{P} \subset \mathcal{X}$, we use the notation $\Delta^t(\mathcal{P}) := \Delta(\mathcal{P}, S_t)$ for brevity.

The total cost can be decomposed as :

$$\Delta^t(\mathcal{X}) = \Delta^t(\mathcal{H}_t) + \Delta^t(\mathcal{U}_t)$$

We can use Lemma B.5 to bound the first term directly.

**Lemma B.6.** *For each $t \in \{1, \ldots, k\}$ the following holds :*

$$\mathbb{E}[\Delta^t(\mathcal{H}_t)] \leq 8\Delta_k(\mathcal{X}) + \frac{2\delta}{1-\delta}\Delta_1(\mathcal{X})$$

*Proof.*

$$\mathbb{E}[\Delta^t(\mathcal{H}_t)] = \mathbb{E}[\sum_{i \in H_t} \Delta^t(\mathcal{C}_i)] \leq \sum_{i=1}^{k} \mathbb{E}[\Delta^t(\mathcal{C}_i)]$$

$$\leq 8\Delta_k(\mathcal{X}) + \frac{\delta}{1-\delta}\mathbb{E}[\Delta^t(\mathcal{X})]$$

$$\leq 8\Delta_k(\mathcal{X}) + \frac{2\delta}{1-\delta}\Delta_1(\mathcal{X})$$

Where in the last line we used Lemma B.3 and the fact that the first center is chosen uniformly at random from $\mathcal{X}$. $\qquad \square$

**Potential function.** To bound the second term i.e, the cost of the uncovered clusters we use the potential function introduced in (Dasgupta, 2013) :

$$\Psi_t = \frac{W_t}{|U_t|}\Delta^t(\mathcal{U}_t)$$

Instead of *paying* the complete clustering cost $\Delta^k(\mathcal{X})$ at once, we make sure that at the end of iteration $t$, we have payed an amount of atleast $\Psi_t$ . Observe that when $t = k$, we have $W_t = |U_t|$ so the potential becomes $\Delta^k(\mathcal{U}_k)$, which is indeed the total cost of the uncovered clusters returned by `RS-k-means++`. We now show how to bound the expected increase in the potential i.e, $\Psi_{t+1} - \Psi_t$. To do this, we shall analyze the error propagation due to using the $D_\delta^2$ distribution instead of the $D^2$ distribution on the way.

**Bounding the Increments.** Suppose $t$ centers have been chosen. The next center $c_{t+1}$ is chosen which belongs to some optimal cluster $\mathcal{C}_i$. We consider two cases : the first case is when $i \in U_t$ i.e, a new cluster is covered and the the second case is when $i \in H_t$ i.e, the center is chosen from an already covered cluster. We shall denote all the set of all random variables after the end of iteration $t$ by $\mathcal{F}_t$.

**Lemma B.7.** *For any $t \in \{1, \ldots, k-1\}$, the following holds :*

$$\mathbb{E}[\Psi_{t+1} - \Psi_t | \mathcal{F}_t, \chi_{U_t}(i)]$$
$$\leq \frac{2\delta}{1-\delta} \frac{t}{\max(1, k-t-1)^2} \Delta_1(\mathcal{X})$$

*Proof.* When $i \in U_t$, we have $W_{t+1} = W_t$, $H_{t+1} = H_t \cup \{i\}$ and $U_{t+1} = U_t \setminus \{i\}$. Thus,

$$\Psi_{t+1} = \frac{W_{t+1}}{|U_{t+1}|} \Delta^{t+1}(\mathcal{U}_{t+1}) \leq \frac{W_t}{|U_t| - 1} \left( \Delta^t(\mathcal{U}_t) - \Delta^t(\mathcal{C}_i) \right)$$

We can use item (ii) of Lemma B.4 for getting a lower bound on the second term :

$$\mathbb{E}[\Delta^t(\mathcal{C}_i) | \mathcal{F}_t, \chi_{U_t}(i)]$$
$$\geq \sum_{j \in U_t} \left( \frac{\Delta^t(\mathcal{C}_j)}{\Delta^t(\mathcal{U}_t)} - \frac{\delta}{1-\delta} \frac{|\mathcal{C}_j|}{|\mathcal{U}_t|} \frac{\Delta^t(\mathcal{C}_j)\Delta^t(\mathcal{X})}{\Delta^t(\mathcal{U}_t)^2} \right) \Delta^t(\mathcal{C}_j)$$
$$\geq \left( 1 - \frac{\delta}{1-\delta} \frac{\Delta^t(\mathcal{X})}{\Delta^t(\mathcal{U}_t)} \right) \sum_{j \in U_t} \frac{\Delta^t(\mathcal{C}_j)^2}{\Delta^t(\mathcal{U}_t)}$$

Where in the second step we used the fact that $|\mathcal{C}_j| \leq |\mathcal{U}_t|$ for each $j \in U_t$. We can use the cauchy-schwarz [19] inequality to simplify the last expression as follows :

$$|U_t|^2 \sum_{j \in U_t} \Delta^t(\mathcal{C}_j)^2 \geq |U_t| \sum_{j \in U_t} \Delta^t(\mathcal{C}_j) = |U_t|\Delta^t(\mathcal{U}_t)$$

This shows that

$$\mathbb{E}[\Delta^t(\mathcal{C}_i) | \mathcal{F}_t, \chi_{U_t}(i)] \geq \frac{\Delta^t(\mathcal{U}_t)}{|U_t|} - \frac{\delta}{1-\delta} \frac{\Delta^t(\mathcal{X})}{|U_t|}$$

Now,

$$\mathbb{E}[\Psi_{t+1} | \mathcal{F}_t, \chi_{U_t}(i)]$$
$$\leq \frac{W_t}{|U_t| - 1} \left( \Delta^t(\mathcal{U}_t) - \mathbb{E}[\Delta^t(\mathcal{C}_i) | \mathcal{F}_t, \chi_{U_t}(i)] \right)$$
$$\leq \frac{W_t}{|U_t| - 1} \left( \Delta^t(\mathcal{U}_t) - \frac{\Delta^t(\mathcal{U}_t)}{|U_t|} + \frac{\delta}{1-\delta} \frac{\Delta^t(\mathcal{X})}{|U_t|} \right)$$
$$= \Psi_t + \frac{\delta}{1-\delta} \frac{W_t}{|U_t| (|U_t| - 1)} \Delta^t(\mathcal{X})$$

Recall that $W_t \leq t$ and $|U_t| \geq k - t$. So for $t \leq k - 2$, the following holds after taking expectation :

$$\mathbb{E}[\Psi_{t+1} - \Psi_t | \mathcal{F}_t, \chi_{U_t}(i)]$$
$$\leq \frac{\delta}{1-\delta} \frac{t}{(k-t-1)^2} \mathbb{E}[\Delta^t(\mathcal{X}) | \mathcal{F}_t, \chi_{U_t}(i)]$$
$$\leq \frac{\delta}{1-\delta} \frac{t}{(k-t-1)^2} \mathbb{E}[\Delta^t(\mathcal{X})]$$
$$\leq \frac{2\delta}{1-\delta} \frac{t}{(k-t-1)^2} \Delta_1(\mathcal{X})$$

---

[19] for lists of numbers $a_1, \ldots, a_m$ and $b_1, \ldots, b_m$ we have $\left( \sum_i a_i b_i \right)^2 \leq \left( \sum_i a_i^2 \right) \left( \sum_i b_i^2 \right)$

Now consider the case when $t = k - 1$. We cannot use the above argument directly because it may so happen that $|U_k| = 0$. If this happens, the potential of the uncovered clusters is always $0$. This only happens when a new cluster is covered in each iteration. Let this event be AC (for *All Clusters* being covered). Denoting $\mathcal{E} = \mathcal{F}_{k-1}, \chi_{U_{k-1}(i)}$ we have the following :

$$\begin{aligned}
\mathbb{E}[\Psi_k - \Psi_{k-1}|\mathcal{E}] &= \mathbb{E}[\Psi_k - \Psi_{k-1}|\mathcal{E}, \text{AC}] \Pr[\text{AC}|\mathcal{E}] \\
&\quad + \mathbb{E}[\Psi_k - \Psi_{k-1}|\mathcal{E}, \neg\text{AC}] \Pr[\neg\text{AC}|\mathcal{E}] \\
&\leq \mathbb{E}[\Psi_k - \Psi_{k-1}|\mathcal{E}, \neg\text{AC}] \Pr[\neg\text{AC}|\mathcal{E}] \\
&\leq \mathbb{E}[\Psi_k - \Psi_{k-1}|\mathcal{E}, \neg\text{AC}] \\
&\leq \frac{2\delta t}{1 - \delta}\Delta_1(\mathcal{X})
\end{aligned}$$

Where in the last line we used the fact that $|U_{k-1}| > 1$ if all clusters are not covered. Combining both cases completes the proof. $\square$

**Lemma B.8.** *For any $t \in \{1, \dots, k - 1\}$, the following holds :*

$$\mathbb{E}[\Psi_{t+1} - \Psi_t|\mathcal{F}_t, \chi_{H_t}(i)] \leq \frac{\Delta^t(\mathcal{U}_t)}{k - t}$$

*Proof.* When $i \in H_t$, we have $H_{t+1} = H_t, W_{t+1} = W_t + 1$ and $U_{t+1} = U_t$. Thus,

$$\begin{aligned}
\Psi_{t+1} - \Psi_t &= \frac{W_{t+1}}{|U_{t+1}|}\Delta^{t+1}(\mathcal{U}_{t+1}) - \frac{W_t}{|U_t|}\Delta^t(\mathcal{U}_t) \\
&\leq \frac{W_t + 1}{|U_t|}\Delta^t(\mathcal{U}_t) - \frac{W_t}{|U_t|}\Delta^t(\mathcal{U}_t) \\
&= \frac{\Delta^t(\mathcal{U}_t)}{|U_t|} \leq \frac{\Delta^t(\mathcal{U}_t)}{k - t}
\end{aligned}$$

$\square$

We can now combine the two cases to get :

**Lemma B.9.** *For any $t \in \{1, \dots, k - 1\}$, the following holds :*

$$\mathbb{E}[\Psi_{t+1} - \Psi_t|\mathcal{F}_t] \leq (1 - \delta)\frac{\mathbb{E}[\Delta^t(\mathcal{H}_t)]}{k - t}$$
$$+ \delta\left(\frac{2}{k - t} + \frac{2t}{\max(1, k - t - 1)^2}\right)\Delta_1(\mathcal{X})$$

*Proof.* To compute the overall expectation, we have :

$$\begin{aligned}
\mathbb{E}[\Psi_{t+1} - \Psi_t|\mathcal{F}_t] &= \mathbb{E}[\Psi_{t+1} - \Psi_t|\mathcal{F}_t, \chi_{U_t}(i)] \Pr[\chi_{U_t}(i)] \\
&\quad + \mathbb{E}[\Psi_{t+1} - \Psi_t|\mathcal{F}_t, \chi_{H_t}(i)] \Pr[\chi_{H_t}(i)]
\end{aligned}$$

We can bound the first term using Lemma B.7

$$\begin{aligned}
&\mathbb{E}[\Psi_{t+1} - \Psi_t|\mathcal{F}_t, \chi_{U_t}(i)] \Pr[\chi_{U_t}(i)] \\
&\leq \mathbb{E}[\Psi_{t+1} - \Psi_t|\mathcal{F}_t, \chi_{U_t}(i)] \\
&\leq \frac{2\delta}{1 - \delta}\frac{t}{\max(1, k - t - 1)^2}\Delta_1(\mathcal{X})
\end{aligned}$$

and the second term using Lemma B.8

$$\begin{aligned}
&\mathbb{E}[\Psi_{t+1} - \Psi_t|\mathcal{F}_t, \chi_{H_t}(i)] \Pr[\chi_{H_t}(i)] \\
&\leq \frac{\Delta^t(\mathcal{U}_t)}{k - t}\left((1 - \delta)\frac{\Delta^t(\mathcal{H}_t)}{\Delta^t(\mathcal{X})} + \delta\frac{|\mathcal{H}_t|}{|\mathcal{X}|}\right) \\
&\leq (1 - \delta)\frac{\Delta^t(\mathcal{H}_t)}{k - t} + \delta\frac{\Delta^t(\mathcal{X})}{k - t}
\end{aligned}$$

Where in the last step we used $\Delta^t(\mathcal{U}_t) \leq \Delta^t(\mathcal{X})$ and $|\mathcal{H}_t| \leq |\mathcal{X}|$. Combining both the terms completes the proof.

$\square$

We are now ready to provide a proof for Theorem B.1, which we state again :

**Theorem B.10.** *Let $\mathcal{X} \subset \mathbb{R}^d$ be any dataset which is to be partitioned into $k$ clusters. Let $S$ be the set of centers returned by $\delta$-`k-means++`$(\mathcal{X}, k, \delta)$ for any $\delta \in (0, 0.5)$. The following approximation guarantee holds :*

$$\mathbb{E}[\Delta(\mathcal{X}, S)] \leq 8(\ln k + 2)\Delta_k(\mathcal{X}) + \frac{6k\delta}{1 - \delta}\Delta_1(\mathcal{X})$$

*Proof.* At the end of $k$ iterations, we have $\Delta(\mathcal{X}, S) = \Delta^t(\mathcal{H}_t) + \Delta^t(\mathcal{U}_t) = \Delta^t(\mathcal{H}_t) + \Psi_k$. The first term can be bound using Lemma B.6. For the second term, we can express $\Psi_k$ as a telescopic sum :

$$\mathbb{E}[\Delta(\mathcal{X}, S)] = \mathbb{E}[\Delta^k(\mathcal{H}_k)] + \sum_{t=0}^{k-1} \mathbb{E}[\Psi_{t+1} - \Psi_t | \mathcal{F}_t]$$

$$\leq \mathbb{E}[\Delta^k(\mathcal{H}_k)] + \sum_{t=0}^{k-1}(1 - \delta)\frac{\mathbb{E}[\Delta^t(H_t)]}{k - t}$$

$$+ \sum_{t=0}^{k-1} \delta\left(\frac{2}{k - t} + \frac{2t}{(1 - \delta)\max(1, k - t - 1)^2}\right)\Delta_1(\mathcal{X})$$

$$\leq 8\Delta_k(\mathcal{X})\left(1 + (1 - \delta)\sum_{t=0}^{k-1}\frac{1}{k - t}\right) +$$

$$\frac{2\delta}{1 - \delta}\Delta_1(\mathcal{X})\left(k + 2\ln k + \sum_{t=0}^{k-2}\frac{t}{(k - t - 1)^2}\right)$$

To simplify this, note that $\sum_{t=0}^{k-1}\frac{1}{k-t} \leq 1 + \ln k$, $\sum_{t=0}^{k-2}\frac{t}{(k-t-1)^2} \leq k\sum_{n=1}^{\infty}n^{-2} = \frac{\pi^2}{6}k$ and $4\ln k \leq \left(4 - \frac{\pi^2}{3}\right)k$ for sufficiently large $k$. Using these above we get our final bound :

$$\mathbb{E}[\Delta(\mathcal{X}, S)] \leq 8(\ln k + 2)\Delta_k(\mathcal{X}) + \frac{6k\delta}{1 - \delta}\Delta_1(\mathcal{X})$$

This completes the proof of the theorem.

$\square$

# C. Experiments

SETUP

All the experiments were performed on a personal laptop with an Apple M3 Pro CPU chip, 11 cores and 18GB RAM. No dimensionality reduction was done on the datasets. No multi - core parallelization was used during the experiments. We have included the code for the experiments in the supplementary material.

DATASETS

The data sets used for the experiments were taken from the annual KDD competitions and the UCI Machine Learning Repository. In the case that the data set consists of a train - test split, only the training data set without the corresponding labels was used for perform clustering. We also provide rough estimates of the $\beta$ parameters for the datasets used. These are computed by taking the ratio of the variance of the dataset with the average clustering cost of the solution output by `RS-k-means++`$(\cdot, \cdot, \infty)$.

*Table 4.* Description of datasets used for experiments

| $\mathcal{X}$ | $n$ | $k$ | $d$ | $\tilde{\beta}_k(\mathcal{X})$ |
|---|---|---|---|---|
| DIABETES (KELLY ET AL., 2021) | $253,680$ | $50$ | $21$ | $\sim 6.5$ |
| FOREST (BLACKARD, 1998) | $581,010$ | $7$ | $54$ | $\sim 3.3$ |
| PROTEIN (CARUANA & JOACHIMS, 2004) | $145,751$ | $100$ | $74$ | $\sim 9.7$ |
| POKER (CATTRAL & OPPACHER, 2002) | $1,025,010$ | $50$ | $10$ | $\sim 2.4$ |
| CANCER (KRISHNAPURAM, 2008) | $94,730$ | $100$ | $117$ | $\sim 1.9$ |

*Table 5.* Comparison of `AF-k-MC`$^2(\cdot, \cdot, 200)$ with `RS-k-means++`$(\cdot, \cdot, \infty)$

| NAME | RS-k-means++ COST | AF-k-MC$^2$ COST | RS-k-means++ STD. DEV. | AF-k-MC$^2$ STD. DEV. | RS-k-means++ TIME | AF-k-MC$^2$ TIME |
|---|---|---|---|---|---|---|
| DIABETES | $7.475 \times 10^6$ | $7.503 \times 10^6$ | $3.23 \times 10^5$ | $3.13 \times 10^5$ | $5.15 \times 10^{-1}$ | $1.02 \times 10^1$ |
| FOREST | $7.707 \times 10^{11}$ | $7.748 \times 10^{11}$ | $1.31 \times 10^{11}$ | $9.61 \times 10^{10}$ | $1.48 \times 10^{-1}$ | $3.51 \times 10^0$ |
| PROTEIN | $2.439 \times 10^{11}$ | $2.436 \times 10^{11}$ | $4.09 \times 10^{10}$ | $4.44 \times 10^{10}$ | $1.06 \times 10^0$ | $1.37 \times 10^1$ |
| POKER | $3.322 \times 10^7$ | $3.333 \times 10^7$ | $5.55 \times 10^5$ | $5.96 \times 10^5$ | $8.95 \times 10^{-1}$ | $5.43 \times 10^1$ |
| CANCER | $6.067 \times 10^6$ | $6.086 \times 10^6$ | $1.19 \times 10^5$ | $7.18 \times 10^4$ | $3.75 \times 10^{-1}$ | $9.69 \times 10^0$ |

ALGORITHMS

1. `RS-k-means++` : Our approach takes as input the parameter $m$ which is an upper bound on the number of iterations of rejection sampling. This provides a trade-off between computational cost and solution quality. We can also set $m = \infty$ to recover the $O(\log k)$ guarantee of `k-means ++`.

2. `AF-k-MC`$^2$ : This is the Monte Carlo Markov Chain based approach of (Bachem et al., 2016a). It also takes as input a parameter $m$ which is the length of the markov chain used for sampling.

*Remark* C.1. We do not include comparisons with the algorithm of (Cohen-Addad et al., 2020) since their techniques are algorithmically sophisticated including tree embeddings and LSH data structures for approximate nearest neighbor search. This incurs additional poly-logarithmic dependence on the aspect ratio of the dataset and even $n^{O(1)}$ terms for performing a single clustering. Moreover, a publicly available implementation is not available to the best of our knowledge. Similar reasons are also mentioned in (Charikar et al., 2023) for not including this algorithm in their experiments as well. As for the algorithm of (Charikar et al., 2023) called `PRONE`, it achieves an $O(k^4 \log k)$ guarantee while running in expected time $O(n \log n)$ after $O(\texttt{nnz}(\mathcal{X}))$ pre-processing. Due to the large approximation factor, (Charikar et al., 2023) suggest to use `PRONE` in a pipeline for constructing coresets instead of clustering the whole dataset. Moreover, the class of datasets targeted by both (Cohen-Addad et al., 2020) and (Charikar et al., 2023) include the large $k(\sim 5 \times 10^3)$ regime, while our approach is more suitable for massive datasets where $n \gg k$. This is because the time taken by our algorithm to perform a single clustering is *sublinear* in $n$, much like the results of (Bachem et al., 2016a). Hence, we compare our approach with their `AF-k-MC`$^2$ algorithm.

**Experiment 1**

In this experiment, we compare the performance of the default `AF-k-MC`$^2$ with $m = 200$ (as done by (Bachem et al., 2016a) in their implementation) with the performance `RS-k-means++` without setting any upper bound for the number of iterations for the datasets given in Table 4. Recall that our algorithm does not require an estimate of $\beta$, thus making it free of any extra parameters which require tuning. The algorithms were run for 20 iterations for computing the averages and standard deviations. We also study the effect of varying the number of clusters $k \in \{5, 10, 20, 50, 100\}$ for each dataset.

**Experiment 2**

In this experiment we study the convergence properties of `RS-k-means++`. We plot the average clustering cost of the solutions output by `RS-k-means++` vs the time taken to compute these solutions and compare these with the baseline `k-means ++`

*Table 6.* Comparison of `RS-k-means++` and `AF-k-MC`$^2$ for different datasets.

| DATASET | $k$ | RS-k-means++ COST | AF-k-MC$^2$ COST | RS-k-means++ STD. DEV. | AF-k-MC$^2$ STD. DEV. | RS-k-means++ TIME | AF-k-MC$^2$ TIME |
|---|---|---|---|---|---|---|---|
| DIABETES | 5 | $2.847 \times 10^7$ | $3.089 \times 10^7$ | $3.59 \times 10^6$ | $4.92 \times 10^6$ | $2.32 \times 10^{-2}$ | $8.71 \times 10^{-1}$ |
| | 10 | $1.768 \times 10^7$ | $1.740 \times 10^7$ | $1.33 \times 10^6$ | $2.23 \times 10^6$ | $4.78 \times 10^{-2}$ | $1.99 \times 10^0$ |
| | 20 | $1.174 \times 10^7$ | $1.195 \times 10^7$ | $5.03 \times 10^5$ | $9.98 \times 10^5$ | $1.32 \times 10^{-1}$ | $4.15 \times 10^0$ |
| | 50 | $7.401 \times 10^6$ | $7.446 \times 10^6$ | $3.26 \times 10^5$ | $2.29 \times 10^5$ | $5.08 \times 10^{-1}$ | $1.03 \times 10^1$ |
| | 100 | $5.515 \times 10^6$ | $5.476 \times 10^6$ | $1.39 \times 10^5$ | $1.25 \times 10^5$ | $1.59 \times 10^0$ | $2.14 \times 10^1$ |
| FOREST | 5 | $1.041 \times 10^{12}$ | $1.062 \times 10^{12}$ | $1.69 \times 10^{11}$ | $1.78 \times 10^{11}$ | $1.13 \times 10^{-1}$ | $2.31 \times 10^0$ |
| | 10 | $5.941 \times 10^{11}$ | $5.853 \times 10^{11}$ | $8.65 \times 10^{10}$ | $6.37 \times 10^{10}$ | $2.47 \times 10^{-1}$ | $5.41 \times 10^0$ |
| | 20 | $3.377 \times 10^{11}$ | $3.373 \times 10^{11}$ | $2.51 \times 10^{10}$ | $2.02 \times 10^{10}$ | $6.97 \times 10^{-1}$ | $1.16 \times 10^1$ |
| | 50 | $1.834 \times 10^{11}$ | $1.846 \times 10^{11}$ | $8.35 \times 10^9$ | $6.48 \times 10^9$ | $3.27 \times 10^0$ | $2.98 \times 10^1$ |
| | 100 | $1.221 \times 10^{11}$ | $1.221 \times 10^{11}$ | $2.64 \times 10^9$ | $3.41 \times 10^9$ | $1.01 \times 10^1$ | $5.85 \times 10^1$ |
| PROTEIN | 5 | $1.048 \times 10^{12}$ | $1.085 \times 10^{12}$ | $2.88 \times 10^{11}$ | $3.00 \times 10^{11}$ | $2.48 \times 10^{-2}$ | $6.21 \times 10^{-1}$ |
| | 10 | $6.394 \times 10^{11}$ | $5.882 \times 10^{11}$ | $9.71 \times 10^{10}$ | $6.15 \times 10^{10}$ | $4.59 \times 10^{-2}$ | $1.40 \times 10^0$ |
| | 20 | $4.388 \times 10^{11}$ | $4.434 \times 10^{11}$ | $2.83 \times 10^{10}$ | $3.81 \times 10^{10}$ | $1.26 \times 10^{-1}$ | $2.93 \times 10^0$ |
| | 50 | $3.029 \times 10^{11}$ | $3.059 \times 10^{11}$ | $1.20 \times 10^{10}$ | $8.55 \times 10^9$ | $3.96 \times 10^{-1}$ | $7.59 \times 10^0$ |
| | 100 | $2.417 \times 10^{11}$ | $2.456 \times 10^{11}$ | $4.73 \times 10^9$ | $5.44 \times 10^9$ | $1.24 \times 10^0$ | $1.47 \times 10^1$ |
| POKER | 5 | $7.81 \times 10^7$ | $8.03 \times 10^7$ | $5.72 \times 10^6$ | $9.10 \times 10^6$ | $4.77 \times 10^{-2}$ | $3.41 \times 10^0$ |
| | 10 | $5.88 \times 10^7$ | $6.04 \times 10^7$ | $2.61 \times 10^6$ | $3.41 \times 10^6$ | $1.14 \times 10^{-1}$ | $8.13 \times 10^0$ |
| | 20 | $4.58 \times 10^7$ | $4.51 \times 10^7$ | $1.70 \times 10^6$ | $1.16 \times 10^6$ | $2.70 \times 10^{-1}$ | $1.64 \times 10^1$ |
| | 50 | $3.31 \times 10^7$ | $3.31 \times 10^7$ | $5.41 \times 10^5$ | $4.68 \times 10^5$ | $8.24 \times 10^{-1}$ | $4.06 \times 10^1$ |
| | 100 | $2.68 \times 10^7$ | $2.69 \times 10^7$ | $4.81 \times 10^5$ | $3.89 \times 10^5$ | $2.07 \times 10^0$ | $8.29 \times 10^1$ |
| CANCER | 5 | $1.21 \times 10^7$ | $1.23 \times 10^7$ | $1.03 \times 10^6$ | $1.17 \times 10^6$ | $1.96 \times 10^{-2}$ | $3.75 \times 10^{-1}$ |
| | 10 | $1.07 \times 10^7$ | $1.06 \times 10^7$ | $5.96 \times 10^5$ | $7.44 \times 10^5$ | $2.46 \times 10^{-2}$ | $8.40 \times 10^{-1}$ |
| | 20 | $8.83 \times 10^6$ | $8.75 \times 10^6$ | $4.02 \times 10^5$ | $4.05 \times 10^5$ | $3.89 \times 10^{-2}$ | $1.84 \times 10^0$ |
| | 50 | $7.02 \times 10^6$ | $7.06 \times 10^6$ | $1.46 \times 10^5$ | $1.96 \times 10^5$ | $8.84 \times 10^{-2}$ | $4.77 \times 10^0$ |
| | 100 | $6.08 \times 10^6$ | $6.06 \times 10^6$ | $1.06 \times 10^5$ | $7.22 \times 10^4$ | $3.69 \times 10^{-1}$ | $9.49 \times 10^0$ |

solution. We also report $95\%$ confidence intervals in the plots over 40 iterations of the algorithms. The plots are generated by varying the upper bound on the number of rejection sampling iterations from $m \in \{5, 10, 20, 50, 75, 100, 125, 150\}$.

**Observations**

Based on the above experiments, we summarize our observations as follows :

- **Observation 1.** The data dependent parameter $\beta$ does not take on prohibitively large values. Indeed, for the data sets used in our experiments, these values are quite reasonable. This observation is in accordance with the experiments of (Bachem et al., 2016b).

- **Observation 2.** `RS-k-means++` provides solutions with comparable quality to `AF-k-MC`$^2$, while generally being much faster. On datasets like POKER where the data size is much larger than the number of clusters, we observe a speedup of $\sim$ 40 - 70$\times$. Moreover, this version of `RS-k-means++` does not require choosing any extra parameters as input.

- **Observation 3.** The solution quality of `RS-k-means++` approaches that of `k-means ++` rapidly with increase in the upper bound for the number of rejection sampling rounds allowed. This can be seen from the plots in Figure 5

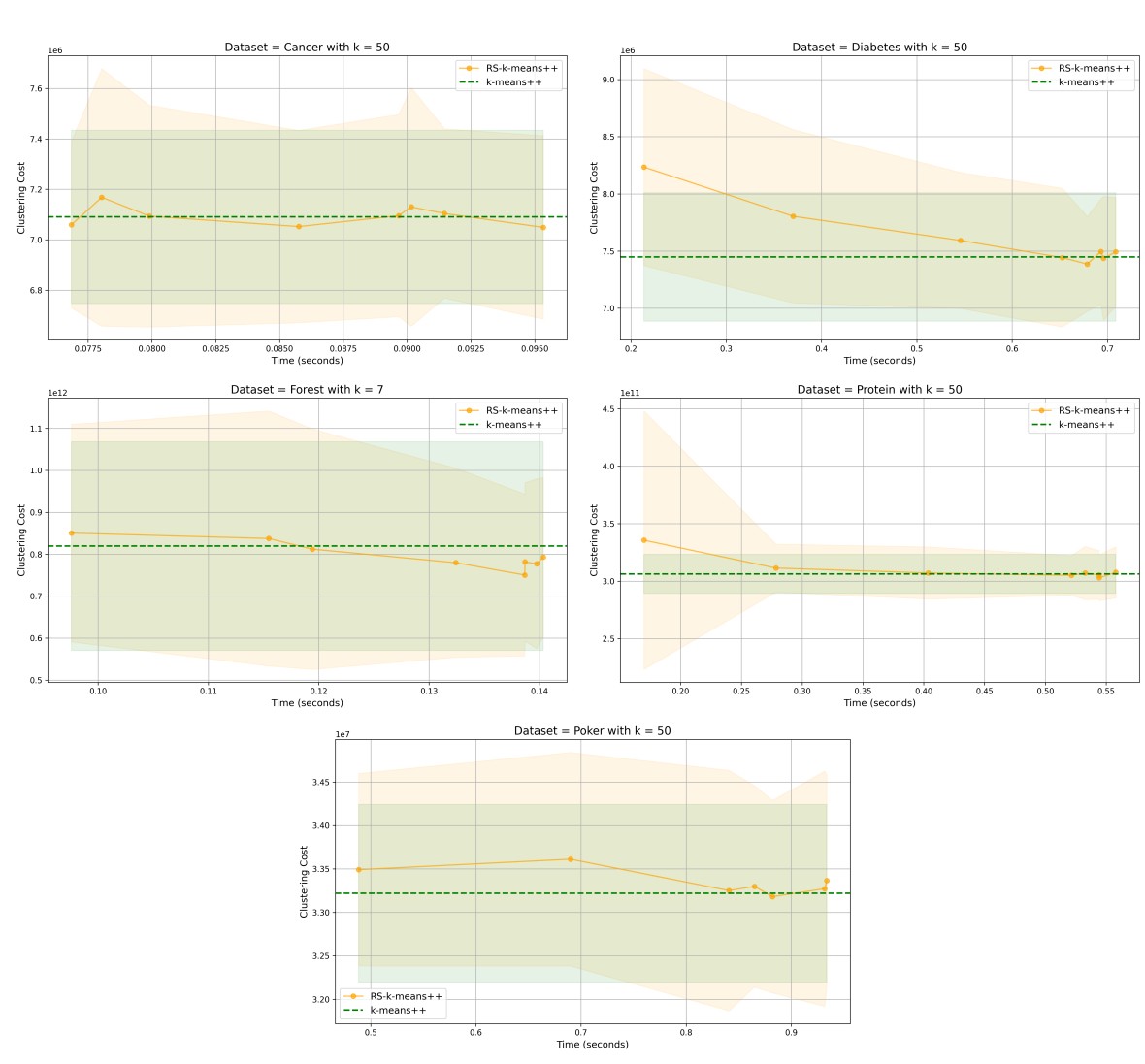

*Figure 3.* Trade-off plots

*Table 7.* Data points for the trade-off plots

| DATASET | $m$ | COST | STD. DEV. | TIME |
|---|---|---|---|---|
| DIABETES | 5 | $8.235 \times 10^6$ | $4.39 \times 10^5$ | $2.14 \times 10^{-1}$ |
| | 10 | $7.804 \times 10^6$ | $3.87 \times 10^5$ | $3.70 \times 10^{-1}$ |
| | 20 | $7.593 \times 10^6$ | $3.04 \times 10^5$ | $5.46 \times 10^{-1}$ |
| | 50 | $7.443 \times 10^6$ | $3.10 \times 10^5$ | $6.53 \times 10^{-1}$ |
| | 75 | $7.495 \times 10^6$ | $2.35 \times 10^5$ | $6.93 \times 10^{-1}$ |
| | 100 | $7.386 \times 10^6$ | $2.11 \times 10^5$ | $6.79 \times 10^{-1}$ |
| | 125 | $7.493 \times 10^6$ | $2.42 \times 10^5$ | $7.09 \times 10^{-1}$ |
| | 150 | $7.437 \times 10^6$ | $2.76 \times 10^5$ | $6.96 \times 10^{-1}$ |
| FOREST | 5 | $8.504 \times 10^{11}$ | $1.32 \times 10^{11}$ | $9.76 \times 10^{-2}$ |
| | 10 | $8.375 \times 10^{11}$ | $1.55 \times 10^{11}$ | $1.15 \times 10^{-1}$ |
| | 20 | $8.122 \times 10^{11}$ | $1.46 \times 10^{11}$ | $1.19 \times 10^{-1}$ |
| | 50 | $7.798 \times 10^{11}$ | $1.15 \times 10^{11}$ | $1.32 \times 10^{-1}$ |
| | 75 | $7.816 \times 10^{11}$ | $9.64 \times 10^{10}$ | $1.39 \times 10^{-1}$ |
| | 100 | $7.504 \times 10^{11}$ | $9.85 \times 10^{10}$ | $1.39 \times 10^{-1}$ |
| | 125 | $7.775 \times 10^{11}$ | $1.03 \times 10^{11}$ | $1.40 \times 10^{-1}$ |
| | 150 | $7.932 \times 10^{11}$ | $9.67 \times 10^{10}$ | $1.40 \times 10^{-1}$ |
| PROTEIN | 5 | $3.356 \times 10^{11}$ | $5.73 \times 10^{10}$ | $1.70 \times 10^{-1}$ |
| | 10 | $3.114 \times 10^{11}$ | $1.06 \times 10^{10}$ | $2.78 \times 10^{-1}$ |
| | 20 | $3.071 \times 10^{11}$ | $1.16 \times 10^{10}$ | $4.04 \times 10^{-1}$ |
| | 50 | $3.070 \times 10^{11}$ | $1.18 \times 10^{10}$ | $5.33 \times 10^{-1}$ |
| | 75 | $3.029 \times 10^{11}$ | $1.01 \times 10^{10}$ | $5.44 \times 10^{-1}$ |
| | 100 | $3.076 \times 10^{11}$ | $1.14 \times 10^{10}$ | $5.58 \times 10^{-1}$ |
| | 125 | $3.054 \times 10^{11}$ | $1.08 \times 10^{10}$ | $5.44 \times 10^{-1}$ |
| | 150 | $3.050 \times 10^{11}$ | $8.80 \times 10^9$ | $5.21 \times 10^{-1}$ |
| POKER | 5 | $3.35 \times 10^7$ | $5.65 \times 10^5$ | $4.88 \times 10^{-1}$ |
| | 10 | $3.36 \times 10^7$ | $6.27 \times 10^5$ | $6.90 \times 10^{-1}$ |
| | 20 | $3.33 \times 10^7$ | $7.06 \times 10^5$ | $8.41 \times 10^{-1}$ |
| | 50 | $3.33 \times 10^7$ | $6.91 \times 10^5$ | $9.32 \times 10^{-1}$ |
| | 75 | $3.33 \times 10^7$ | $5.91 \times 10^5$ | $8.65 \times 10^{-1}$ |
| | 100 | $3.32 \times 10^7$ | $5.65 \times 10^5$ | $8.82 \times 10^{-1}$ |
| | 125 | $3.34 \times 10^7$ | $6.21 \times 10^5$ | $9.34 \times 10^{-1}$ |
| | 150 | $3.34 \times 10^7$ | $6.21 \times 10^5$ | $9.34 \times 10^{-1}$ |
| CANCER | 5 | $7.17 \times 10^6$ | $2.60 \times 10^5$ | $7.80 \times 10^{-2}$ |
| | 10 | $7.06 \times 10^6$ | $1.68 \times 10^5$ | $7.68 \times 10^{-2}$ |
| | 20 | $7.05 \times 10^6$ | $1.95 \times 10^5$ | $8.58 \times 10^{-2}$ |
| | 50 | $7.05 \times 10^6$ | $1.85 \times 10^5$ | $9.53 \times 10^{-2}$ |
| | 75 | $7.13 \times 10^6$ | $2.41 \times 10^5$ | $9.02 \times 10^{-2}$ |
| | 100 | $7.10 \times 10^6$ | $1.71 \times 10^5$ | $9.15 \times 10^{-2}$ |
| | 125 | $7.09 \times 10^6$ | $2.24 \times 10^5$ | $7.99 \times 10^{-2}$ |
| | 150 | $7.10 \times 10^6$ | $2.04 \times 10^5$ | $8.97 \times 10^{-2}$ |

