# OpenReview forum: "A New Rejection Sampling  Approach to  $k\text{-}\mathtt{means}$++ with Improved Tradeoffs"
_ICML.cc/2025/Conference — Submitted to ICML 2025_

### Official Review · Reviewer_yGTT · 2025-03-07

**Overall Recommendation:** 3

**Summary:**

This paper presents a new fast adaptation of k-means++ algorithm.

**Claims And Evidence:**

The algorithm and results are clearly stated and match the claims in the paper.

The only claim that is not supported is the advantages of the approach in this work compared to the baselines. Authors provide comparison with both Bachman and Cohen-Added results but it is not clear at all if they outperform any of them (the running time presented in this paper for Cohen-addad seems to be incorrect, see Cor 5.5).

**Essential References Not Discussed:**

No.

**Experimental Designs Or Analyses:**

Yes. The experimental section is in a good shape, various databases are considered and the results are conviencing.

**Methods And Evaluation Criteria:**

Yes.

**Other Comments Or Suggestions:**

No

**Other Strengths And Weaknesses:**

Strengths:
- The paper is well-written.
- The experimental section is convincing.
- The problem studied is important, and improving the running time has been one of the main challenges in this field.

Weaknesses:
- The theoretical advantages of this work are not clear.
- The novelty is below the expectation for this conference.

**Questions For Authors:**

Please address the concern above.

**Relation To Broader Scientific Literature:**

This paper is interesting in the field of k-means clustering.

**Theoretical Claims:**

It is not clear from the paper in what regimes their algorithm outperforms the baselines. The theoretical results seems to be correct.

---

> ### Author Rebuttal · Authors · 2025-03-31
>
> Dear reviewer yGTT,
>
> Thank you for your thoughtful and constructive review. We address the concerns below:
>
> > (1) The running time presented in this paper for Cohen-Addad seems to be incorrect ...
>
>  The statement in Cor 5.5 did not explicitly state the dependence on $k$ for the algorithm, so we decided to use the more explicit statement in section E.2. Our notation differs slightly in using $\eta \leftrightarrow \Delta$ and $\varepsilon^{-1} \leftrightarrow c^2$.
>
>
> > (2) The only claim that is not supported is the advantages of the approach in this work compared to the baselines ...
>
> The advantages of MCMC methods over the standard kmeans++ are well studied in the literature ([1],[2]), in being much faster while providing competitive seeding quality. So we use the MCMC method as our _"updated"_ baseline and design our experiments to show the advantages of our method over [2]. This is because as shown in [2], the $O(|X|kd)$ run-time of the standard kmeans++ quickly becomes prohibitive for large datasets. However, if the reviewer feels that including the comparison with standard  kmeans++  will increase the readability of our paper, then we will be sure to include it in the final version.
>
> > (3) ... What regimes their algorithm outperforms the baselines ...
>
> As discussed in response (5) to reviewer X1mD, it is possible to construct cases where the standard kmeans++ outperforms both MCMC and our method. This corresponds to datasets with large $\beta$ values. In the small $\beta$ value regime (which we observed for several large datasets), both MCMC and our method outperform kmeans++. Within this regime, our method (RS-kmeans++$(\cdot,\cdot,\infty)$) will consistently outperform MCMC since it adjusts automatically to the $\beta$ values as opposed to MCMC, where an appropriate parameter dependent on $\beta$ must be provided. We also have another method where we provide a similar parameter as input, but even here we obtain provably better convergence properties as compared to the MCMC method.
>
> > (4) The theoretical advantages of this work are not clear $\dots$
>
> We highlight our key contributions :
>
> 1._New trade-offs_. We present a method of performing kmeans++ seeding in time $\tilde{O}(nnz(X) + mk^2d)$ with the approximation guarantee having an additive and scale invariant $k^{-\Omega(m/\beta)} \operatorname{Var}(X)$. Note that this has an exponentially decaying dependence on $m$, improving upon previously known MCMC methods which attain $m^{-1} \operatorname{Var}(X)$ additive error (which is linearly decaying).
>
> 2._Automatic determination of the correct number of iterations_. Our approach which runs in time $O(\beta k^2d)$ does not require a pre-specification of the number of iterations of rejection sampling. Indeed, RS-kmeans++ is \emph{``smart"} in the sense that it will spend less on datasets that are not well-clusterable as opposed to kmeans++, which runs in the same time regardless of the dataset.
>
> 3._Theoretical Analysis of $\delta$-kmeans++_. We use a careful potential-based argument to bound the solution quality of $\delta$-kmeans++, which was not previously known. We believe that this may be of independent interest as well.
>
>
> 4._Fast data updates_. The particularly simple form of the probability distribution that we sample from can be exploited to support pre-processing in input sparsity and data updates in $O(\log |X|)$ time, which may be useful in scenarios where there are dynamic updates to the data being clustered.
>
> 5._Parallel setting_. Unlike the inherently sequential MCMC methods, our rejection sampling approach can be easily parallelized since each rejection sampling step is independent of others.
>
>
>
> We shall be glad to address any other concerns the reviewer may have.

---

### Official Review · Reviewer_f83S · 2025-03-13

**Overall Recommendation:** 2

**Summary:**

The paper proposed a new seeding algorithm for kmeans clustering, accelerating the kmeans++ by leveraging rejection sampling techniques. The key idea is to select new centroids in a way that maintains probabilistic separation from existing ones (alike the kmeans++). Instead of explicitly computing the D2 distribution, which requires evaluating all data points, the authors introduce a data structure that explosit feature sparsity to enable efficient sampling. They proved that their method remains O(log k)-competitive, matching the theoretical guarantee of k-means++, while achieving up to 70x speed up over a prior MCMC-based seeding method.

**Claims And Evidence:**

The primary claims are supported by both theoretical analysis and empirical results. For minor problems, see below.

**Essential References Not Discussed:**

Lastest progresses seem to have been included.

**Experimental Designs Or Analyses:**

The analysis is rigorous, and the experiments are well-structured to support the authors’ primary claims. However, further clarification on the practical impact of the proposed method, particularly in the context of overall k-means efficiency, would strengthen the paper.

**Methods And Evaluation Criteria:**

However, I have concerns regarding the practical applicability of the proposed methods, particularly in two aspects:

1. Limited impact on overall clustering efficiency.

Since the proposed algorithm optimizes only the seeding phase of clustering, its contribution to the total runtime improvement may be marginal. The primary computational cost in k-means stems from the iterative refinement steps, not the initialization. Therefore, the claimed speedup might be overstated. The authors should provide stronger justification for why accelerating the seeding step translates to meaningful performance gains in practical applications.

2. Dependence on data sparsity.

The method appears to leverage feature sparsity to achieve its efficiency gains. However, many modern applications, such as image and text embeddings, involve dense vector representations. How does the proposed approach generalize to such settings? The authors should clarify its effectiveness in scenarios where data sparsity cannot be exploited.

**Other Comments Or Suggestions:**

There are a few ambiguous notations and possible typos:

1. Page 6, line 279 (RHS): Should $\lVert X \rVert$ be $\lVert \mathcal{X} \rVert $? This notation inconsistency appears in multiple places, including the appendix. The authors should clarify if this is a typo or an intended distinction.

2. Notation system clarity. For instance:
- The difference between $\lVert \mathcal{X} \rVert$ and  $|\mathcal{X}|$  is unclear without prior explanation.
- The meaning of  nnz($mathcal{X}$)  (number of nonzero entries in $\mathcal{X}$ should be explicitly stated. $\mathcal{X}$ is a set, while nnz($mathcal{X}$) seems to denote the non-zero elements in the feature matrix.

**Other Strengths And Weaknesses:**

The introduced adaptive trade-off between computational cost and solution quality is intriguing and has the potential for broader impact beyond k-means++ seeding. If similar techniques can be applied to other clustering or optimization problems, this approach could be valuable in balancing efficiency and accuracy in large-scale machine learning tasks.

**Questions For Authors:**

1. Can the Lloyd iteration be accelerated with the proposed technique?

**Relation To Broader Scientific Literature:**

The contribution to the broader scientific literature is questionable, as the work focuses on a highly specific topic—accelerating a particular seeding algorithm used only in the initialization phase of k-means clustering. Given that k-means iterations typically dominate the overall computational cost, the impact of this improvement on general clustering applications remains unclear. The authors should better position their work within the broader context of clustering and machine learning to justify its significance beyond this niche problem.

**Theoretical Claims:**

The proofs appear to be correct under a basic check. However, I have some concerns:

1. Accounting for preprocessing time in total complexity

The claimed time complexity does not seem to explicitly include the cost of the preprocessing step. In particular, the centering step (subtracting the mean from all points) inherently requires  $O(|\mathcal{X}| d)$ operations, regardless of data sparsity. Since this centering operation is a necessary preprocessing step for the proposed method—but not for prior approaches—its cost should be accounted for in the overall complexity analysis. However, it appears to be omitted from the provided bounds.

---

> ### Author Rebuttal · Authors · 2025-03-31
>
> Dear Reviewer f83S
>
> Thank you for your thoughtful and constructive review. We address the concerns below:
>
> > (1) Limited impact on ...
>
>  There are many reasons to speed up the $D^2$-sampling-based seeding itself, and this has already been well addressed in the literature, for example (paragraph 3, introduction) of [1]. The reasons can be summarized as follows :
>
> 1.Extending the inherently sequential kmeans++ to the distributed setting is non-trivial while Lloyd's iterations is easily adapted in the MapReduce framework.
>
> 2.Many use cases require just the seeding - including components of clustering algorithms in the online / streaming / distributed setting.
>
> 3.Many theoretical results and practical algorithms for coreset constructions etc incorporate  kmeans++ (as pointed out by reviewer 1rwo as well).
>
> Any speedup in kmeans++ seeding percolates to all of these downstream applications.
>
> > (2) Dependence on data sparsity.
>
> 1.We emphasize that the key advantage of our method is not due to using the sparsity present in the data but due to using  rejection sampling for sampling from the $D^2$ distribution instead of computing it wholly. This also allows us to prove improved trade-offs.
>
> 2.The pre-processing time is proportional to the time it takes to just read the input, and the main loop for performing the seeding requires just ${O}(mk^2d)$ operations, which is independent of $|X|$.
>
> 3.If the data is indeed sparse and is given in a sparse representation, then our method is able to exploit that sparsity for faster pre-processing.
>
> 4.In the experiments as well, the speedups reported are solely due to the complexity of the main loop. We shall make this more explicit in the final version.
>
>
> > (3) Accounting for preprocessing time in total complexity
>
> Thank you very much for pointing this out. The way the pseudocode is written currently indeed takes $O(|X|d)$ time but can easily be done in $O(nnz(X))$ time as well. Hence, the time complexity mentioned for our method includes the pre-processing time as well. We will be sure to explicitly discuss this in the final version. We now describe how this can be done:
>
> Let the dataset be $X = \{x_1,\dots,x_n\}$. The mean of the dataset can be computed in $O(nnz (X))$ time as follows: We initialize $\mu$ to be the all-zero vector and upon seeing an entry $(i,j,v)$ perform the update $\mu(i) \gets \mu(i) + v$ and normalize afterwards. We also compute the norm $\|\mu\|$ in $O(d)$ time.  This takes $nnz(X)+d$ updates. Now, to enable sampling from $D_X$, we need to compute the norm for each point of $X$. For each $x \in X$, we can compute $\|x\|^2$ in $nnz(x)$ operations and $\braket{x,\mu}$ in $nnz(x)$ operations as well, since $\sum_{i=1}^d x(i) \mu(i) = \sum_{i : x(i) \neq 0} x(i)\mu(i)$. Since $\|x - \mu\|^2 = \|x\|^2 + \|\mu\|^2 - 2\braket{x,\mu}$, we can compute all the norms in $\sum_{i=1}^nO(nnz(x_i)) = O(nnz(X))$ operations as well. In the sampling step, we compute the distance evaluations using the already computed mean $\mu$, so the complexity of the main loop remains $\tilde{O}(mk^2d)$. So, we need not explicitly center the dataset in $O(|X|d)$ time. We assume that the dataset is centered to make our proofs more simple.
>
> > (4) The authors should clarify whether  ...
>
> The sample and query data structures are not necessary for implementing our method if the data is not sparse or dynamic updates are not required. The reason for including them is to show how the particularly simple form of the probability distribution that we sample from can be exploited to support pre-processing in input sparsity and data updates in $O(\log |X|)$ time, which may be useful in scenarios where there are dynamic updates to the data being clustered, in which case one does not have to re-run the pre-processing step. We leave the choice of implementing them as a design choice since the key speed-up is due to the main loop.
>
> > (5) Notation Issues
>
> $\|X\|$ denotes the frobenius norm $\|X\| = \sqrt{\sum_{x \in X} \|x\|^2}$, while $|X|$ denotes the number of data points in the dataset $X$. For a vector $x \in \mathbb{R}^d$, $\mathtt{nnz}(x)$ denotes the number of non-zero components of $x$. For the dataset $X$, we use $nnz(X) = \sum_{x \in X} nnz(x)X$. $\|X\|$ should be $\|X\|$ instead. (For the rebuttal we use $X$ instead due to space constraints)
>
> > (6) Can the Lloyd iteration ...
>
>  No, our method specifically speeds up the seeding step of kmeans++, which, as discussed, is a useful case in itself for designing clustering algorithms. Since Lloyd's iterations are very sensitive to the initial centers chosen, the seeding step is important in practice to decrease the number of iterations required for convergence.
>
> We hope that we were able to address all concerns. We shall be glad to provide any further clarifications.
>
> [1] : Bachem, O., Lucic, M., Hassani, S. H., and Krause,
> A. Approximate k-means++ in sublinear time. AAAI 2016

---

### Official Review · Reviewer_X1mD · 2025-03-14

**Overall Recommendation:** 3

**Summary:**

The paper introduces rejection sampling, an alternative sampling method of k-means++ initialization. By making approximations, the method is able to make faster initial k clusters than its similar counterpart that uses MCMC based sampling on tasks where the number of datapoints is substantially higher than the number of clusters.

The method seems exciting. But the paper needs some work in regards to presentation (see below).

**Claims And Evidence:**

The claims have extensive theoretical analysis and are verified with experiments.

**Essential References Not Discussed:**

None.

**Experimental Designs Or Analyses:**

The designs are valid. However, the plot comparison with the original k-means++ should also be done with the MCMC method. Also, while they claim to show two methods, the comparison is only done on the second proposed sampling method.

**Methods And Evaluation Criteria:**

The datasets are adequate for their method comparison.

**Other Comments Or Suggestions:**

Figure A3 seems to be a duplicate of Figure 1, just with different order of the plots. Figure A3 also has a seemingly unintended black line. Same citation style is used for both in sentence citation and out of sentence citation, confusing the readers. The major concepts used in the Introduction are later described in the Preliminary section, which is an obstacle to reading the paper sequentially.

Figure 1 has incredibly small fonts.

The code is hard to run in its current state. Assumes Linux operating system, although the code is written in Python which is cross-platform.

Table 7 not referenced in the document.

**Other Strengths And Weaknesses:**

The strength of the paper is a mathematically sound yet simple way of approximating k-means++. K-means++ is often a time bottleneck when running multiple runs of k-means and this could be easily incorporated into open source packages such as scikit-learn or matlab codes to quickly test out its effectiveness.

Weakness is, though reasons are explained in the appendix, the amount of comparison results. We can see that it provides faster results than the MCMC based method mentioned and compared against, but it is hard to convince readers to try the new method out in place of what they already have, despite the simplicity of the method.

More end-to-end comparisons are needed with RS-k-means -> k-means are needed with reports of NMI, ARI and ACC that many readers are familiar with.

**Questions For Authors:**

Are there datasets that will be troublesome for RS-k-means++ but not for k-means++?
For example datasets with only a few samples?
Please elaborate.

**Relation To Broader Scientific Literature:**

The key contributions suggest a fast yet simple way of approximating k-means++. This could be very useful in scenarios where one is testing for the optimum number of clusters and have to run multiple runs of k-means on a single dataset with large number of points (e.g. brain function MRI data).

**Theoretical Claims:**

Yes I have checked the mathematical proofs on why the two sampling methods can replicate the k-means++ initialization with certain approximation guarantee.

---

> ### Author Rebuttal · Authors · 2025-03-31
>
> Dear reviewer X1mD,
>
> We thank you for your thoughtful and constructive review and suggestions. We address the concerns below:
>
> > (1) The plot comparison ... should also be done with the MCMC method.
>
> As far as the convergence properties are concerned, the MCMC method converges to kmeans++ as well, and that is why we felt that it was sufficient to compare the solution cost computed by the standard kmeans++ method. However, we now think that it will be more informative to include the plot for MCMC, as suggested by the reviewer, in the final version as well.
>
> > (2) while they claim  ... the comparison is only done on the second proposed sampling method.
>
> The RS-kmeans++ algorithm takes the parameter value $m$, which is the upper bound on the number of rejection sampling iterations allowed. This parameter controls the solution quality v/s computational cost trade-off. The two methods that we propose differ only on the setting of $m$. RS-kmeans++$(\cdot,\cdot, \infty)$ doesn't put a bound on the number of rejection sampling iterations and has the same $O(\log k)$ guarantee as kmeans++, while RS-kmeans++$(\cdot,\cdot,m)$ allows the user to control the trade-off.
>
> Experiment 1 shows the effectiveness of the approach when no upper bound is placed on the number of rejection sampling iterations. Due to the strong convergence guarantees which we prove, the behavior of RS-kmeans++$(\cdot, \cdot,m)$ approaches that of RS-kmeans++$(\cdot,\cdot,\infty)$. Experiment 2 is designed to test this, and it is observed that this convergence happens even for small values of $m$. Hence, we believe that the experiments are sufficient to show the overall effectiveness of RS-kmeans++.
>
> > (3) It provides faster results than the MCMC ...  but it is hard to convince readers to try the new method ...  despite the simplicity of the method.
>
>  We have integrated a basic implementation of our algorithm as a Python library, which can be used along with popular packages like NumPy and Scikit-learn. We hope that this would help the community to easily try out our methods: https://anonymous.4open.science/r/RSkmeanspp-D240/. This integration allows users to try out our method instead of the MCMC method by simply changing a single line of code : `centers = rskmeanspp(X, k, m)` in place of `centers = afkmc2(X, k, m)`. Hence, we do not clearly see why someone would be less likely to try out our faster methods. Moreover, our methods have the added advantages of being able to do the pre-processing in input sparsity, supporting fast data updates and being easy to extend to the parallel setting.
>
> > (4) More end-to-end comparisons are needed ... with reports of NMI, ARI and ACC ...
>
> We emphasize that RS-kmeans++ is just a faster _implementation_ of kmeans++ seeding, the properties of which are well studied. If the reviewer still feels that including the reports of NMI/ARI/ACC would improve the readability of the paper, then we would be glad to include the reports or citations to relevant literature in the final version.
>
>
>
> > (5) Are there datasets that will be troublesome for RS-k-means++ but not for k-means++?  ...
>
> It is possible to construct a pathological dataset that theoretically has a very large value of the $\beta$ parameter. For example, [1] constructs such an example where the inter-cluster distance is $\delta$ and intra-cluster distance is $\Delta$ where $\Delta \gg \delta$. Due to a large $\beta$ value, the theoretical runtime is slower than kmeans++ if $\beta \simeq \Delta/\delta \in  \Omega(n/k)$. RS-kmeans++ can be thought of as a _''smart"_ algorithm, which spends less time on datasets that are not well-clusterable.  However, we see that the $\beta$ values for real-life datasets are quite reasonable.  Indeed, even for datasets with $n/k \sim 10^5$, we observe $\beta \sim 10$.
>
> Our method targets datasets that are very large, i.e, $n = |X| \gg k$, since then the runtime of the classic implementation of kmeans++ i.e, $O(nkd)$ becomes prohibitively large, while our main loop has only logarithmic dependence on $n$ i.e., $\tilde{O}(mk^2d)$
>
> > (6) ...  Assumes Linux operating system, although the code is written in Python, which is cross-platform ...
>
> The anonymized Python library pointed to in an earlier comment should take care of this.
>
> > (7) Figure A3 seems to be a duplicate of Figure 1 ...
>
> We included the plots so that the experimental section in the appendix is complete by itself.
>
> > (8) Presentation Issues
>
> We thank the reviewer for their suggestions and shall make sure that these are incorporated in the final version.

---

### Official Review · Reviewer_1rwo · 2025-03-15

**Overall Recommendation:** 3

**Summary:**

The paper gives a faster version for the $k$-means++ algorithm while maintaining the approximation guarantee offered by the original $k$-means++. They try to approximate the $D^2$ sampling used in $k$-means++. The authors first preprocess the data to center it and make a data structure that allows sampling from the centered data based on its norm. Now they sample data points from a distribution which is a weighted combination of $D^2$ and uniform distribution. The authors prove that their algorithm achieves same guarantees in less time. They empirically demonstrate the effectiveness of their algorithm on real datasets.

**Claims And Evidence:**

The claims in the paper appear valid to me. Paper is clearly written for the most parts.

**Essential References Not Discussed:**

There is a paper "Scalable $k$-means clustering via Light Weight Coresets" by Bachem et al. which also uses a distribution very similar to the one mentioned in this paper. It should be compared and contrasted with. The familiarity with that paper makes me question the novelty of this paper a bit.

**Experimental Designs Or Analyses:**

Seems good.

**Methods And Evaluation Criteria:**

yes

**Other Comments Or Suggestions:**

See the responses to other sections

**Other Strengths And Weaknesses:**

See the responses to other sections

**Questions For Authors:**

Please address the comparison with the paper mentioned in Essential References

**Relation To Broader Scientific Literature:**

$k$-means++ is a very popular algorithm in Machine learning. It is an essential ingredient of many other algorithms like coreset construction algorithms. This paper tries to speed up the algorithm while keeping its approximation guarantee. It would be of interest to the community.

**Theoretical Claims:**

I did not check the proofs in detail but at high level they look alright.

---

> ### Author Rebuttal · Authors · 2025-03-31
>
> Dear Reviewer 1rwo,
> > (1) Please address the comparison with the paper ...
>
> We thank the reviewer for pointing out [1] which we had missed out. Most importantly, we point out the difference in the distributions used by us and [1]. [1] uses the distribution $q(x) = 1/2|X| + \Delta(x,\mu)/2\Delta(X,\mu)$ where $µ$
> is the data mean to construct a coreset of size  $O(\epsilon^{-2}kd \log k)$ , while we use a _sequence_ of distributions
> given by $p(x) = (1−\delta)\Delta(x,S)/\Delta(X,S) + \delta/|X|$ (where $S$ is the currently sampled centers) to sample precisely k centers. Although the distributions are different, let us discuss the results of [1] since we missed that in our submission :
>
> [1] introduced the notion of _lightweight coresets (LWC)_, where the classical definition of a coreset is relaxed to allow for an additive error.  It is shown that if $m = O(\epsilon^{-2}kd\log k)$ points are chosen with probability $q(x) := 1/2|X| + \Delta(x,\mu(X))/2\Delta(X, \mu(X))$ with weight $w(x) = 1/mq(x)$ to form a LWC $C$, then for any set of centers $S$, the following holds:
> $$ E[\Delta(C,S)] \leq  \Delta(X,S) + 4{\epsilon} \Delta(X,\mu(X))  $$
>
> To perform seeding, [1] first constructs a LWC $C$ and, second, applies the standard kmeans++ method to $C$ instead of $X$ (Section 8 of [1]). The total tun time is hence $$O(nnz(X)) + O(mkd) = O(nnz(X)) + O(\epsilon^{-2} k^2d^2 \log k)$$ Combining the above result with the $O(\log k)$  guarantee of kmeans++ gives the following, where $\Delta$ is the cost incurred by the LWC method :
>
>
> $$ E[\Delta] \leq  8 (\ln k +2) \Delta_k(X) + 4{\epsilon} \Delta_1(X) $$
>
> Note that this is a weaker theoretical result as compared to the MCMC method of [2], where the same guarantee is obtained in time $O(nnz(X)) + O(\epsilon^{-1} k^2d)$ (as compared to $O(\epsilon^{-2}k^2d^2 \log k)$ of LWC).
>
> Since our approach improves upon [2], we have an improved trade-off as compared with [1] as well. Note that the goal of [1] is the _construction of LWCs_ in the distributed setting (the first part of their algorithm), while our goal is to speed up kmeans++ itself (the second part of their method). To obtain the seeding, the LWC method has to apply _some_ kmeans method (the kmeans++ algorithm in this case ) to the constructed coreset.
>
> Let us see some key elements of our work and [1] :
>
> 1. As discussed above, the LWC method for kmeans++ attains an approximation guarantee of $O(\log k) \Delta_k + \varepsilon \Delta_1$ in time $O(nnz(X)) + O(\epsilon^{-2}k^2d^2\log k)$, while our method attains $O(\log k) \Delta_k + k^{-\Omega(1 /\varepsilon \beta)} \Delta_1$ in time $O(nnz(X))+O(\varepsilon^{-1}k^2d)$.
>
> 2. Time complexity sublinear in  $|X|$ after pre-processing is achieved in [1] by constructing an LWC, while we achieve this by using rejection sampling to sample from the $D^2$ distribution.
>
> 3. The trade-off in $[1]$ arises due to larger coresets being a better approximation for the complete dataset, while our trade-off arises due to the fact that more number of rejection sampling iterations are a better approximation to the actual $D^2$ distribution.
>
> 4. To quantify this advantage, we need to analyze the solution quality of a variant of kmeans++ which we call $\delta$-kmeans++, where the next center is sampled with probability $(1-\delta) \frac{\Delta(x,S)}{\Delta(X,S)}+ \delta \frac{1}{|X|}$, and precisely $k$ centers are sampled. Here, $\delta$ can be understood as the _``failure probability"_ i.e., the probability that all samples are rejected in $m$ iterations, in which case we sample a point uniformly at random. We use a careful potential-based argument to bound the error propagation in the solution quality due to using the weighted distribution. We were not able to find the required properties in any of the previous works, including [1], even though similar sampling distributions were being used.
>
>
>
>
> Hence, we believe that our contributions are conceptually and technically different from [1].
>
> An advantage of $[1]$ is that the sampling step in the coreset construction can be implemented via a distributed protocol. We point out that our clustering algorithm can be extended to this setting as well. For simplicity, assume that we have a central node, and the dataset $X$ is distributed between $\ell$ nodes as $X_1,X_2,\dots, X_\ell$. It suffices to show how to sample from $D_X$.  Each node $j$ computes  $\|X_j\|$ and makes this available to the central node. The central node computes $\|X\|$.
> To sample a point from $D_X$, the central node chooses a node $j$ with probability $\|X_j\|^2/\|X\|^2$. Node $j$ then samples from $D_{X_j}$ and returns the point to the central node. The central node can then perform the rejection sampling procedure.
>
>
>
> [1]: Scalable and Distributed Clustering via Lightweight Coresets.
> Olivier Bachem, Mario Lucic and Andreas Krause. KDD 2018.
>
> [2]: Fast and provably good seedings for k-means.  Bachem, O., Lucic, M., Hassani, H., and Krause, A. . NIPS 2016

---

### Decision · Program_Chairs · 2025-05-01

**Decision:**

Reject

**Comment:**

This paper utilizes rejection sampling to slightly improve the popular k-means++ seeding algorithm. My own opinion is that it is not clear this is preferable to existing methods for k-means++ and its accelerations, but the attempt at analyzing a rejection sampler and a version that trades off approximation error for speedup because the first version is not particularly competitive contains interesting ideas. The reviews are quite brief with no discussion but lean toward a weak accept. After discussion with reviewers and chairs, this borderline paper may benefit from the constructive criticism outlined by reviewers in a stronger revision, but ultimately did not meet as high a bar as other very competitive papers in this year's batch